

# Monitoring multiple satellite Aerosol Optical Depth (AOD) products within the Copernicus Atmosphere Monitoring Service (CAMS) data assimilation system

Sebastien Garrigues[1], Samuel Remy[2], Julien Chimot[3] , Melanie Ades[1], Antje Inness[1],  Johannes
Flemming[1] , Zak Kipling[1], Istvan laszlo[4] , Angela Benedetti[1], Roberto Ribas[1] ,  Soheila
Jafariserajehlou[3],  Bertrand Fougnie[3], Shobha Kondragunta[4],  Richard Engelen[1],Vincent-Henri Peuch[1],
Mark Parrington[1],  Nicolas Bousserez[1], Margarita Vazquez Navarro[3] , Anna Agusti-Panareda[1]

1: ECMWF, Reading, RG2 9AX, UK
2: HYGEOS, Lille, France
3: EUMETSAT, Darmstadt, 64295, Germany
4: Center for Satellite Applications and Research, NOAA/NESDIS,
College Park, USA

*Correspondence to*: Sebastien Garrigues (sebastien.garrigues@ecmwf.int)





**Abstract**. The Copernicus Atmosphere Monitoring Service (CAMS) provides near real time forecast and reanalysis of aerosols using the ECMWF Integrated Forecasting System with atmospheric composition extension, constrained by the assimilation of MODIS and PMAp Aerosol Optical Depth (AOD). The objective of this work is to evaluate two new near real time AOD products to prepare their assimilation in CAMS, namely the Copernicus AOD from SLSTR (collection 1) on board Sentinel 3-A&B over ocean and the NOAA EPS AOD (v2.r1) from VIIRS on board S-NPP and NOAA-20 over both land and ocean.

The differences between MODIS (C6.1), PMAp (v2.1), VIIRS (v2.r1) and SLSTR (C1) AOD as well as their departure with the modelled AOD were assessed at the model grid resolution (i.e. level-3), using 3-month AOD average (December 2019 - February 2020 and March - May 2020).

VIIRS and MODIS show the best consistency across the products, which is explained by instrument and retrieval algorithm similarities. VIIRS AOD is frequently lower over the ocean background and higher over biomass burning and dust source land
regions compared to MODIS. VIIRS shows larger spatial coverage over land and resolves finer spatial structures such as the transport of Australian biomass burning smoke over the Pacific which can be explained by the higher spatial resolution of the VIIRS level-2 product and the use of a heavy aerosol detection test in the retrieval algorithm. Our results confirm the positive offset over ocean i) between TERRA/MODIS and AQUA/MODIS due to the non-corrected radiometric calibration degradation of TERRA/MODIS in the dark target algorithm and ii) between SNPP/VIIRS and NOAA20/VIIRS due to the positive bias in
the solar reflective bands of SNPP/VIIRS. SLSTR AOD shows much smaller level-3 values than the rest of the products which is mainly related to differences in spatial representativity at the IFS grid spatial resolution due to the stringent cloud filtering applied to the SLSTR radiances. Finally, the geometry characteristics of the instrument, which drive the range of scattering angles sampled by the instrument, can explain a large part of the differences between retrievals such as the positive offset between PMAp data sets from METOP-B and METOP-A.





## 1 Introduction

While aerosol models generally capture the global spatial distribution of aerosols, they can show large differences in aerosol
mass budgets at both global and regional scales (Schugens et al., 2020., Bellouin et al., 2020; Glib et al., 2021). Sessions et al.
(2015) showed that models frequently overestimate low AOD in case of fine size particles and underestimate large AOD in
case of high aerosol load. Uncertainties in the representation of aerosol processes are related to the large horizontal, vertical
and temporal variability of aerosol physicochemical properties (e.g. size, shape, optical properties); the large range of natural
and anthropogenic emission sources, the complex emission, deposition and aging processes which are strongly coupled with
meteorology (e.g. transport, impact of humidity, convective dust storm) and chemistry (e.g. heterogeneous chemistry) (Remy
et al., 2019, Ryder et al., 2019, Burgos et al., 2020, Sand et al., 2021; Glib et al., 2021). Reducing these sources of uncertainties
represent a crucial challenge to improve the representation of aerosol-climate interactions which has been identified as one of
the research priorities for the development of the next generation of Earth system model (IPCC 2021).

Aerosol optical depth (AOD) observation, which measures the extinction of light by aerosols from the surface to the top of the
atmosphere at a given spectral band, is frequently used to constraint global aerosol forecast system through data assimilation.
Satellite AOD observations offer a great potential to resolve the horizontal, vertical and temporal distribution of aerosols
(Kokhanovsky and de Leeuw, 2009; Dubovik et al., 2019). The first satellite AOD data sets were not accurate enough to be
assimilated in global aerosol models due to incomplete spatial coverage, coarse spatial resolution (~ 1°), limited spectral
information content, uncertainties in cloud detection and radiometric calibration. With the advent of enhanced information
content from more recent satellite instruments, more accurate aerosol retrieval data sets have been produced from SEAWIFs
(Sayer et al., 2012), MERIS (Vidot et al., 2008), AATSR (North et al., 1999), MISR (Witek et al., 2013), POLDER (Tanre et
al., 2012), MODIS (Levy et al., 2013) and VIIRS (Sayer et al., 2017; Hsu et al., 2019). Most global aerosol data assimilation
systems have been relying on MODIS which has been providing AOD and fine-mode aerosol fraction over land and ocean
since 2001. Positive impacts for aerosol near real time (NRT) predictions have been shown by Benedetti et al, 2009 for the
European Centre for Medium-Range Weather Forecasts (ECMWF) Integrated Forecast System (IFS) and Zhang et al., 2008,
2014 for the Naval Research Laboratory (NRL) Aerosol Analysis and Prediction System (NAAPS). Besides, Xian et al., 2019
showed that AOD data assimilation improves the consistency between model predictions.

However, aerosol retrieval is a challenging task due to the weak aerosol signal that needs to be separated out from the larger
cloud and surface reflectances and can be affected by various sources of uncertainties. While the Global Climate Observing
System (GCOS) requirement for satellite AOD uncertainty is the larger of 0.03 or 10% of AOD, the uncertainty of most AOD
satellite products falls between 0.02 and 0.05 AOD. It is frequently larger over land than ocean because of land surface
brightness and anisotropy (Sayer et al., 2020). While AOD products generally agree on global average and show consistent
temporal variations, they can substantially differ at regional scale (Kinne, 2009, de leeuw, et al., 2015, Sogacheva, 2020,
Shugens 2020). The analysis of the diversity between AOD products brings additional information on their uncertainty by



identifying the retrieval configurations and the surface types which generate the largest differences between products (Schutgens et al, 2020).

The Copernicus Atmosphere Monitoring Service (CAMS) provides global reanalysis records (Inness et al., 2019) and 5-day global forecasts of greenhouse gases (GHG), trace gas species (Flemming et al., 2015, Huijnen et al., 2019) and aerosols (Remy et al., 2019). CAMS relies on the use of the IFS, which combines state-of-the-art meteorological and atmospheric

composition models together with a 4D-VAR data assimilation scheme. For aerosols, AOD at 550 nm derived from MODIS and PMAp (synergy between GOME-2, IASI, AVHRR on board Metop-A, B, C) datasets are operationally assimilated in CAMS. Implementing new observational data streams is a priority for CAMS to increase the spatial and temporal coverage of the assimilated observations, to increase the resilience of the data assimilation system to instrument failures and to enhance the accuracy of the analysis. It requires two preparatory steps. First, new satellite observations must be properly evaluated at

the model spatial resolution to check their consistency with both the modelled AOD and the other satellite AOD used in the data assimilation system, at both regional and global scales. Since biases in the observations and departure between observations can significantly affect the data assimilation outputs (Zhang and Reid, 2005), the spatial and temporal structure of the systematic differences between satellite products need to be properly understood and quantified in order to account for them in the assimilation process (Dee et al., 2005). The second step consists in implementing and testing the assimilation of

the new products, which includes adapting the bias correction scheme and evaluating the observation error. The present work focuses on the passive monitoring of new satellite AOD data sets (first step) and the impact of their assimilation (second step) will be addressed in a separate paper.

While the uncertainty of satellite AOD products and their diversity have been documented in various studies (Sogacheva et al., 2020, Schutgens et al., 2020, 2021), the evaluation was frequently done at the native spatial and temporal resolution of the

retrieval (denoted hereafter level-2). No recent studies have evaluated AOD products within a data assimilation system for NRT applications. While some AOD products such as MODIS or the VIIRS NASA product have been extensively evaluated (Sogacheva et al., 2020; Schutgens et al., 2021), the PMAp data set, the recent Copernicus NRT SLSTR product and the NOAA EPS VIIRS product have not been intercompared. Besides, most existing intercomparison exercises compared collocated level-2 satellite retrievals and evaluated them against independent ground measurements such as AERONET. While

level-2 accuracy and uncertainty information was probably enough when assimilating a single AOD product, the current challenge to design efficient multi-satellite AOD assimilation strategies is to better understand the diversity of level-3 AOD products at the model grid spatial resolution.

The objective of this work is to evaluate two new AOD products to prepare their future assimilation in CAMS, namely the Copernicus NRT AOD product (collection 1) from SLSTR on board Sentinel 3-A&B over ocean and the NOAA EPS AOD

product (v2.r1) from VIIRS on board S-NPP and NOAA-20 over both land and ocean. The consistency between MODIS (C6.1), PMAp (v2.1), VIIRS (v2.r1) and SLSTR (C1) AOD products as well as their differences with the modelled AOD were monitored over a 6-month experiment, from December 2019 to May 2020. This paper aims at assessing the differences between the satellite AOD products at the IFS model grid resolution (i.e. level-3). All analyses and conclusions reported in this paper



hold for level-3 satellite AOD generated for their use in the CAMS data assimilation system and may not directly apply to
level-2 retrieval. Multi-month AOD averages were compared to characterize the systematic differences between products. The
first guess departure, which represents the differences between the observation and the model state variable prior to the
assimilation, is a key metric operationally used by NWP centres to characterize the systematic and the random errors between
the observation and the model (Bell et al., 2008) and prepare the implementation of new satellite observations (Rennie et al.,
2021). It is used in this work to identify possible inconsistencies between the investigated AOD products within the CAMS
data assimilation system.

Section 2 provides a description of the satellite AOD observations used in this work. Section 3 presents the IFS model used in
CAMS, the simulation experiments designed for this work and the intercomparison methodology. The results are summarized
in Sect. 4. The main sources of differences between the investigated AOD products are discussed in Sect. 5. Conclusions and
recommendations from this work are given in Sect. 6.




**Table 1: Characteristics of the satellite AOD products**

| Product | Instrument / Platform ( overpass time) | Version | Spatial resolution | Period | Retrieval algorithm | Uncertainty (EE) |
|---|---|---|---|---|---|---|
| MOD04_L2<br><br>MYDO4_L2 | MODIS/TERRA (10.00 am)<br>MODIS/AQUA (1.30pm) | C6.1 | 10km | From 2000 | **Land:**<br><br>- Dark Target (DT) on vegetated surfaces: spectral relationships<br><br>- Deep Blue (DB) on vegetated and bright surfaces: spectral relationships and surface reflectance database<br><br>**Ocean**: ocean surface reflectance model +LUT for 6 spectral bands | DT: Empirical expression from evaluation against AERONET, EE=+/-(0.05+15% $AOD_{AERONET}$)<br><br>DB: Empirical expression from evaluation against AERONET, EE=+/- (a+b*$AOD_{MODIS}$)/ (1/cos($\theta_0$ )+ 1/ cos($\theta$)), $\theta_0$ and $\theta$ are the view and solar angles, the coefficients a and b can be found at https://atmosphere-imager.gsfc.nasa.gov/sites/default/files/ModAtmo/modis_deep_blue_c61_changes2.pdf |
| NOAA EPS AOD | VIIRS/SNPP, NOAA20 (1-1.30pm) | V2r1 | 0.750 km | From 2012 for SNPP and 2018 for NOAA 20 | **Ocean:** ocean surface reflectance model +LUT for 7 spectral bands | Empirical expression from evaluation against AERONET (separate land and ocean parametrization), methodology described in Huang et al., (2016) |





| | | | | | **Land:**<br>- vegetated surfaces: spectral relationships<br>- bright surfaces: spectral relationships and surface reflectance database | |
|---|---|---|---|---|---|---|
| COPERNICUS SLSTR NRT AOD | SLSTR/S3A, S3B (10.00 am) | C1 | 9.5km | From August 2020 | **Ocean:** ocean surface reflectance model + inversion using nadir and oblique views<br><br>**Land:** joint aerosol-surface retrieval from dual-view model and spectral constraints | Prognostic (from the optimization algorithm) |
| PMAp | GOME-2, IASI, AVHRR /METOP-A,B,C (9.30 am) | V2.1 | PMAp-A:40*5km$^2$ PMAp-B: 40*10km$^2$ | Ocean: from 2014<br><br>Land: from 2016 | Multi-sensor Algorithm, AOD retrieved from GOME-2 PMD bands, distinct LUTs over land and ocean | Prognostic (from the optimization algorithm) |




## 2 Satellite AOD products

The satellite products investigated in this work are the MODIS AOD C6.1 from TERRA and AQUA produced by NASA (Levy et al., 2013; Hsu et al., 2019), the NOAA EPS AOD v2.r1 from VIIRS on board S-NPP and NOAA-20 produced by NOAA (Laszlo and Liu, 2020), the Copernicus NRT AOD C1 from SLSTR on board Sentinel 3-A&B produced by

EUMETSAT with the Optimized Simultaneous Surface Aerosol Retrieval for Copernicus Sentinel-3 (OSSAR-CS3) (Chimot et al., 2021) and PMAp (v2.1) derived from GOME-2, IASI and AVHRR instruments on board Metop-A, B, C produced by EUMETSAT (Grzegorski et al., 2022). All these products provide AOD at 0.55 μm for clear-sky and daylight conditions. Below we describe their general characteristics, which are summarized in Table 1. More detailed descriptions along with validation statements can be found in Appendix A.

## 2.1 AOD product characteristics

### 2.1.1 MODIS and VIIRS

MODIS and VIIRS AOD are derived from two imaging radiometers which have similar spectral information contents. MODIS AOD has a ~10 km spatial resolution while VIIRS is retrieved at the native spatial resolution of the VIIRS radiances (0.750 km). MODIS product includes two distinct retrieval algorithms: the dark target (DT) over dark surfaces (ocean, vegetated

areas) and the deep blue (DB) over dark and bright land surfaces.

Over ocean, the MODIS DT and the VIIRS algorithm have similar characteristics. The ocean surface reflectance is calculated from an ocean surface reflectance model, which represents the contributions from sun-glint, underwater and whitecap reflections. They exploit similar fine and coarse mode aerosol models adopted from Remer et al., 2006.

Over vegetated land surfaces, MODIS (DT and DB) and VIIRS algorithms exploit a similar spectral constraint approach which

consists in estimating the surface reflectance in the visible from the SWIR (or the RED and the SWIR for VIIRS) which is assumed to be slightly affected by atmospheric scattering (Kaufman et al., 1997, Levy et al., 2013., Hsu et al., 2013, Lazlo and Liu, 2020). Over bright and heterogeneous surfaces, both the MODIS DB and the VIIRS algorithms exploit a surface reflectance database to represent the surface anisotropy, the surface spatial variability and the seasonal changes of the surface reflective properties. MODIS DT uses a combination of a dust model with non-spherical shape and one of three fine-mode

aerosol models with spherical shape and different absorbing properties. MODIS DB exploit 10 fine-mode and 5 coarse-mode aerosol models with spherical shape but conversely to DT a single aerosol model is selected for the optimal solution. 4 aerosol models, which are essentially based on the Collection 5 MODIS DT models, are used by the VIIRS algorithm which dynamically selects the aerosol model based on the value of the residual between calculated and observed reflectances.




### 2.1.2 SLSTR

Conversely to MODIS and VIIRS, SLSTR has dual-view capability with a nadir and an oblique view pointing backward at 55°. The AOD product is provided at a spatial resolution of 9.5 km. Over ocean, the retrieval relies on the spectral information content from all available views which are used as independent spectral observations. The surface reflectance is pre-calculated

using an ocean BRDF model, which includes contributions from glint, white foam and ocean colour and uses the wind speed from the ECMWF forecast. Over land, the retrieval algorithm is a combination of the North et al., 1999 dual-angular model, used in a joint aerosol-surface reflectance fit, and a spectral first guess for the RED surface reflectance derived from the NIR or the SWIR radiances (Chimot et al., 2021).

### 2.1.3 PMAp

Conversely to the other products based on single instrument, PMAp is derived from the synergistic use of the GOME-2 UV-VIS spectrometer, the IASI Fourier transform infrared sounding interferometer and the AVHRR radiometer on board METOP platforms. The TOA reflectances derived from the GOME-2 Polarisation Measurement Devices (PMDs) measurements are the main inputs of the AOD retrieval while AVHRR and IASI observations are exploited for aerosol type identification and cloud detection (Grzegorski et al., 2022). PMAp has a much coarser spatial resolution (5 x 40 km$^2$ for Metop-A and 10 x 40

km$^2$ for Metop-B) than the rest of the products.

### 2.2 Implementation in CAMS

While MODIS and VIIRS retrievals are exploited over both land and ocean, only ocean retrievals are considered for PMAp and SLSTR because land retrievals of PMAP v2.1 and SLSTR C.1 were deemed to not be accurate enough for their assimilation in CAMS. In CAMS, MODIS DT retrievals associated with a quality assessment (QA) equal to three over land and larger or

equal to one over ocean are selected. DB retrievals associated with QA larger or equal to two are used to gap-fill DT over land. Best quality retrievals are selected for VIIRS, SLSTR and PMAp.

## 3 Model and experiment design

### 3.1 IFS Model

CAMS relies on the use of the Integrated Forecasting System (IFS), which is the NWP model developed at ECMWF. IFS includes state-of-the-art atmospheric transport, chemistry (Flemming et al., 2015, Huijnen et al., 2019) and aerosol (Remy et al., 2019) models and is constrained by a 4D-VAR data assimilation scheme. CAMS produces 5-day forecasts and reanalysis of aerosol, reactive and greenhouse gases. In the CAMS operational configuration, the simulations are performed at the horizontal spectral resolution of TL511 (equivalent to a grid size of about 40 km) and a vertical resolution of 137 levels (0.01

to 1013 hpa). The IFS Cycle 47R1 was used in this work and the full documentation can be found at





https://www.ecmwf.int/en/publications/ifs-documentation (last access: 9 May 2021). We provide below the main characteristics of the atmospheric composition modelling components of the IFS and the 4DVAR data assimilation scheme.

### 3.1.1 Atmospheric Transport

Advection of the atmospheric tracers is simulated by a semi-Lagrangian scheme (Temperton et al. 2001). A mass fixer has
been implemented to ensure conservation of mass of atmospheric species during transport (Agustí-Panareda et al., 2014). Vertical mixing is simulated from the IFS turbulent diffusion and convection schemes.

### 3.1.2 Chemistry

Chemistry is represented using a modified version of the Carbon Bond 05 model (CB05, Yarwood et al., 2015) for the troposphere. More detailed on the IFS-CB05 system can be found in Flemming et al. (2015) and Huijnen et al. (2016,2019).
### 3.1.3 Aerosol

Aerosol mass mixing ratios are simulated using a bulk-bin scheme (Boucher et al., 2002). 14 species are represented which includes 3 size bins for dust and sea salt (defined at 80% humidity), hydrophilic and hydrophobic organic matter and black carbon, sulphate, ammonium, fine mode nitrate produced from gas-particle partitioning and coarse-mode nitrate produced from heterogeneous reactions. Emission of sea-salt and dust as well as the conversion of sulphur dioxide into sulphate and
nitrate are computed online using the IFS meteorological variables. The Global Fire Assimilation System (GFASv1.4), provides globally gridded hourly estimates of biomass burning emission fluxes for reactive gas, greenhouse gases and aerosols based on assimilated MODIS observations of Fire Radiative Power (FRP) (Kaiser et al., 2012). The rest of the static primary aerosol sources are provided by the CAMS-GLOB-ANT 4.2 emission inventory data set for anthropogenic sources (Elguindi et al., 2020) and from the CAMS-GLOB-BIO v1.1 emissions inventory, based on the MEGAN
model (Sindelarova et al., 2014) with ERA-Interim reanalysis meteorology, for biogenic sources. The emission of secondary organic aerosol is scaled on anthropogenic CO emissions and is added to organic matter emissions (Remy et al., 2019). More detailed on aerosol modelling can be found in Remy et al. (2019, 2021).

### 3.1.4 Data assimilation

Meteorology and atmospheric composition control variables are initialized using the incremental 4D-VAR assimilation scheme
implemented in IFS (Courtier et al., 1994). In order to reduce the computational cost and the impact of model non-linearities, the minimization is achieved at a lower spatial resolution and using simplified physics (only atmospheric transport is represented for aerosols and chemistry). The assimilation is performed twice a day over a 12-hour assimilation window. For aerosols, MODIS C6.1 is assimilated over ocean and land and PMAp v2.1 is assimilated over ocean only. A thinning at 0.5-degree spatial resolution is applied to both MODIS and PMAp to reduce the amount of observations and minimize the impacts
of horizontal correlation on the observation error. A variational bias correction scheme (Dee et al, 2004) is applied to PMAp and MODIS is used to anchor the bias correction, i.e. not bias corrected. The aerosol data assimilation scheme is further described in Benedetti et al. (2009).



### 3.2 CAMS AOD performances

The evaluation of the CAMS cycle 47R1 against AERONET shows a positive bias at global scale which is higher over North America and an underestimation of dust but with large regional variability (e.g. overestimation over Sahara and underestimation in the Sahel and the Mediterranean region) (Schulz et al., 2020). The burden of fine mode aerosols, in particular that of sulfate, appears to be generally too high.

### 3.3 Experiment design

IFS was run from 01/12/2019 to 30/05/2020 using the CAMS operational configuration. The experiments were initialized from a past experiment with a similar configuration. VIIRS and SLSTR AOD were passively monitored while MODIS and PMAp AOD were assimilated. For VIIRS, a superobbing (Janjić et al., 2018) at the TL511 model spatial resolution was applied to reduce the number of observations and comply with the IFS computing requirements while preserving the main spatial patterns resolved by the VIIRS product. No spatial thinning was applied to SLSTR observations because the product is distributed at a coarser spatial resolution than the native spatial resolution of the retrieval and the stringent cloud filtering applied to the input radiances used in the retrieval algorithm leads to a substantial reduction of the number of AOD observations.

### 3.3 Intercomparison methodology

The goal of this work is to assess the differences between the level-3 AOD satellite products generated within the CAMS data assimilation system and their departures with the model. Two 3-month periods, namely December 2019-February 2020 (DJF) and March 2020-May 2020 (MAM), were distinguished in the evaluation. For the DJF period, the 16 and 17 of January data are discarded for all the products because the VIIRS AOD product was affected by calibration errors in the VIIRS reflectances on these two days. The intercomparsison was carried out at the IFS model spatial resolution (~40 km) and at a 3-month temporal resolution. This was done in two steps: (1) instantaneous regridding of level-2 retrieval product at the (level-3) IFS model spatial resolution and (2) 3-month average of the instantaneous level-3 AOD retrieval. The intercomparison of the satellite products at the model spatial resolution, which is much coarser than the level-2 retrieval spatial resolution, should reduce the impacts of the differences in spatial resolution between products (Sayer al, 2019). Besides, the use of a multi-month AOD average should minimize the impacts of differences in temporal representativity (Schutgens et al., 2017) and allows us to focus on the systematic differences between products.

Observations above 70°N and below 70°S were disregarded because they generally do not meet the quality criteria for their assimilation. The product comparison was carried out over ocean and land separately. Over land, it includes MODIS and VIIRS while over ocean it includes MODIS, VIIRS, SLSTR and PMAp. Distinct regional domains were defined over land and ocean to encompass a large range of aerosol characteristics and surface types for which the retrieval algorithms may exhibit different behaviours. These domains are defined in Table 2 and 3 for ocean and land, respectively.



Differences between the temporal averages of the products were assessed through global and regional maps, product versus

product scatter plots and latitudinal transects. They were quantified by the mean deviation (MD), the root mean square of the differences (RMSD) and the Pearson correlation coefficient (r). The samples used to compute these metrics were the spatial and 3-month AOD average within each model grid box.

A key metric, used in this work, is the first guess departure which represents the differences between the observation and the value of the modelled AOD (first guess) prior to the assimilation of MODIS and PMAp AOD. It was computed by mapping

the modelled AOD in the observation space independently for each satellite product which consists in first interpolating the simulated aerosol mixing ratios to the observation location and time, and then computing the modelled AOD from the mixing ratio value using the aerosol observation operator (Benedetti et al., 2009). The first guess departure represents the differences between the level-2 retrieval at its native spatial resolution and the model-simulated equivalent observation with collocation in time and space. It should be noted that the first guess also includes the impact of assimilated MODIS and PMAp AOD from

the previous cycles. Geographical maps of the mean and the standard deviation of the first guess departure in space and time are produced by taking the observation-model collocated samples within each model grid box and within each 3-month period. Given the large number of spatial and temporal samples used, the mean and the standard deviation (SD) of the first guess departure are meaningful estimates of the systematic and random, respectively, differences between the observation and the model that can be used to identify possible spatial inconsistencies between AOD satellite products.




**Table 2: Definion of the regions of interest over ocean**

| Domain names | Latitude N | Latitude S | Longitude W | Longitude E |
|---|---|---|---|---|
| Mid-Atlantic (MA) | 20°N | 10 °S | 55°W | 20°E |
| North Atlantic (NA) | 55°N | 30°N | 60°W | 20°W |
| North Pacific (NP) | 55°N | 30°N | 175°W | 30°W |
| South Pacific (SP) | 20°S | 55°S | 175°W | 80°W |
| South Indian ocean (SIO) | 10°S | 60°S | 35°E | 160°E |
| Arabic peninsula and India coast (AI) | 25°N | 10°S | 25°E | 90°E |


**Table 3: Definion of the regions of interest over land**

| Domain names | Latitude N | Latitude S | Longitude W | Longitude E |
|---|---|---|---|---|
| Africa (AF) | 35°N | 35°S | 30°W | 60°E |
| Asia (AS) | 66°N | 10°S | 60°E | 180°E |
| Australia (AU) | 10°S | 50°S | 100°E | 180°E |
| Europe (EU) | 66°N | 35°N | 30°W | 60°E |
| North-America (NA) | 66°N | 15°N | 170°W | 50°W |
| South-America (SA) | 15°N | 60°S | 95°W | 30°W |



## 4 Results

### 4.1 Evaluation over ocean

### 4.1.1 Satellite observations

Figure 1 and 2 indicate that PMAp has the highest global mean AOD. The global mean of SLSTR is half that of the rest of the products. VIIRS global means are 0.01 and 0.02 lower than MODIS global means for DJF and MAM, respectively. PMAP shows the largest spatial variability (SD=0.1) while the rest of the products have similar global SD (~0.07). Over the remote ocean, SLSTR shows much lower AOD than the rest of the products and its global mean is half that of MODIS. VIIRS exhibits spatial structures of low AOD over the North Pacific and the North Atlantic for the DJF period (Fig. 1) and over the South ocean for the MAM period (Fig. 2). These structures are smaller and noisier in the MODIS and PMAp maps and they cannot be distinguished in the SLSTR maps. Products exhibit large diversity in the South ocean where PMAp and MODIS show nosier spatial patterns compared to VIIRS and SLSTR. All products consistently show high AOD values in the tropical Atlantic (Fig. 1), resulting from Sahara dust transport and smoke from central African biomass burning areas, and off the India and China coasts (Fig. 2) due to continental aerosol transport. VIIRS and SLSTR both detect the 2019-2020 Australian fire smoke transport over the Pacific (20°S to 50°S, 120°W to 180°W) where PMAp and MODIS show noisier spatial patterns (Fig. 1). The aerosol plumes detected by SLSTR over ocean have a smaller extent and are more fragmented compared to the rest of the products.

The global probability distribution functions (PDF) of AOD are displayed for the DJF period in Fig. 3 (results are similar for the MAM period and thus not shown). SLSTR has the narrowest PDF, positively skewed and centred over smaller values compared to the rest of the products. PMAp shows the widest PDF. MODIS and VIIRS exhibit similar Gaussian-like distributions but the VIIRS distribution is shifted toward lower values.

Figure 4 confirms the negative departure between SLSTR and the rest of the products across latitudes for the DJF period. PMAp-B shows the highest values while PMAp-A is in better agreement with TERRA/MODIS. S3A/SLSTR and S3B/SLSTR have consistent AOD latitudinal transects. SNPP/VIIRS is frequently lower than TERRA/MODIS and is closer to AQUA/MODIS. NOAA-20/VIIRS is lower than SNPP/VIIRS and the differences are larger in the Southern Hemisphere. MODIS, VIIRS and PMAp-A show larger discrepancies over south mid-latitudes (20°S - 60°S) compared to north mid-latitudes and the tropics. All the products display a similar AOD peak at ~2 °N to 8°N, which is related to the frequent Atlantic dust outbreak for which MODIS has higher values than VIIRS. The increase of SLSTR and VIIRS AOD around 30°S is related to the Australian fire smoke transport shown in the global maps. The increase of SLSTR AOD above 55°N is related to artefacts in the retrieval at high latitude. Similar differences in AOD latitude transects are obtained for the MAM period presented in Fig. D.1.

The values of MD, RMSD and r between products are reported in Table 4 and the associated scatterplots are given in appendix B.1. VIIRS and MODIS have the smallest MD, RMSD and the highest correlation. The absolute MD between VIIRS and MODIS is slightly larger when using VIIRS from NOAA20 than from SNPP. SLSTR shows large negative MD with both





VIIRS (-0.06) and MODIS (-0.08). PMAp has the smallest correlation with MODIS compared to SLSTR and VIIRS. RMSD
and MD are lower between VIIRS and MODIS/AQUA compared to VIIRS and MODIS/TERRA. Similar differences between
products are obtained for the MAM period (results not shown) with slightly higher MD between VIIRS and MODIS.

Table 5 and Fig C.1 characterize the diversity between products at the regional scale. Wider AOD distributions are reported in
the South ocean compared to global scale. SLSTR exhibits larger negative MD in the South ocean compared to the North
Atlantic and the North Pacific domains where MD and RMSD between products are frequently the smallest.


**Figure 1: Global maps of temporal mean AOD from TERRA&AQUA/MODIS, NOAA20&SNPP/VIIRS, S3A&S3B/SLSTR, METOP-A&B/PMAp for the DJF (2019-2020) period over ocean.**






**Figure 2: Global maps of temporal mean AOD from TERRA&AQUA/MODIS, NOAA20&SNPP/VIIRS, S3A&S3B/SLSTR, Metop-A&B/PMAp, for the MAM (2020) period, over ocean.**







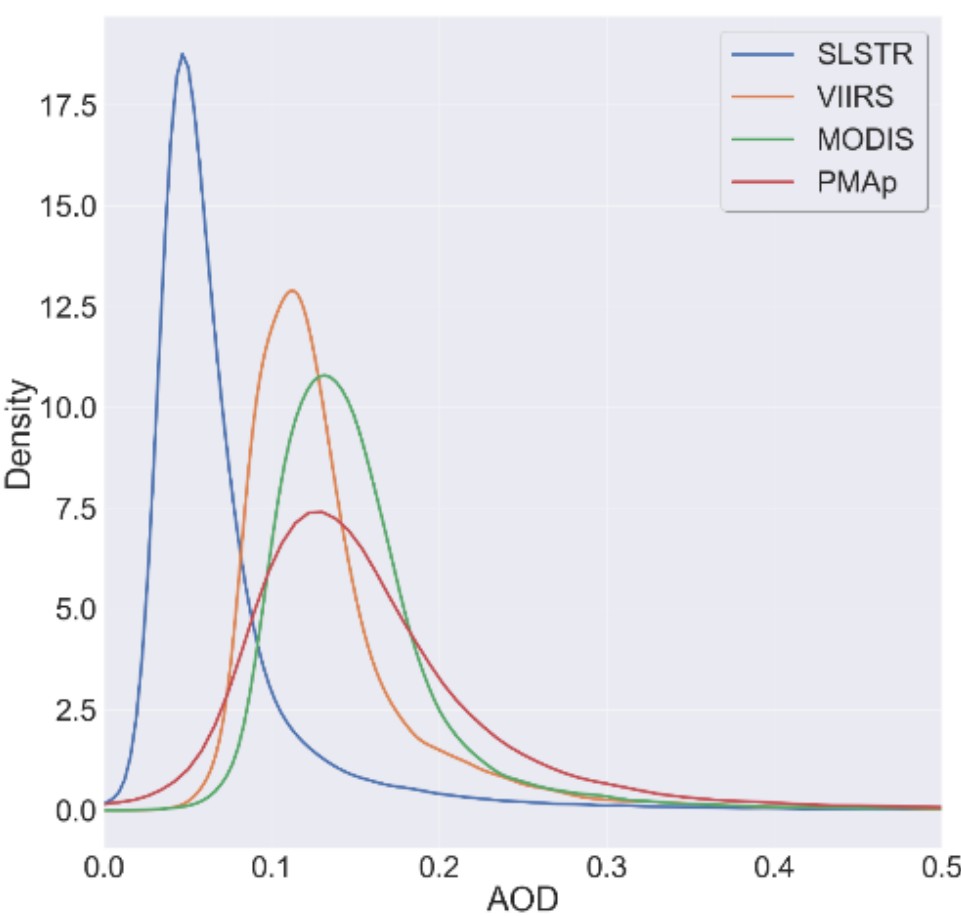


**Figure 3: Global distributions of satellite AOD over ocean for the DJF (2019-2020) period.**






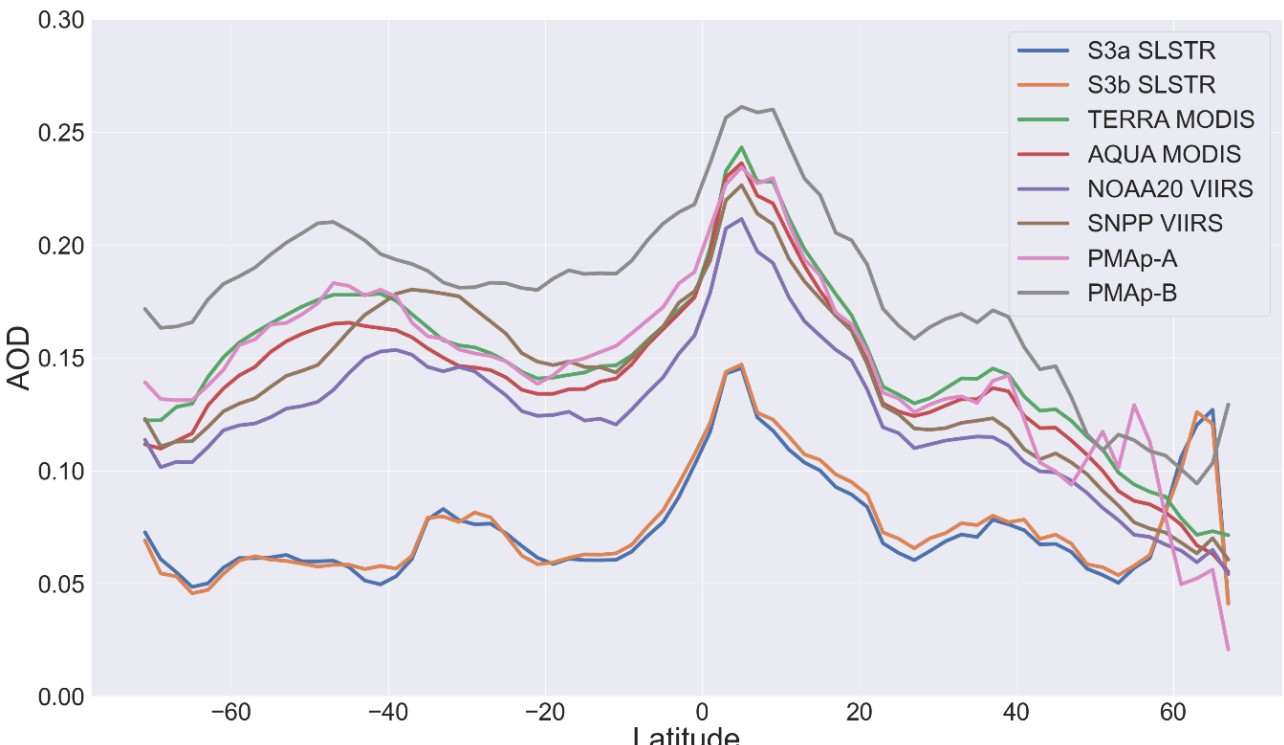

Figure 4: Latitude cross-section of temporal mean satellite AOD, for the DJF (2019-2020) period, over ocean.




**Table 4: Quantification of global AOD differences between satellite and instruments for the DJF period over ocean**

| | RMSD | MD | r |
|---|---|---|---|
| **SLSTR vs TERRA/MODIS** | 0.1 | -0.09 | 0.74 |
| **SLSTR vs AQUA/MODIS** | 0.09 | -0.08 | 0.76 |
| **SLSTR vs MODIS** | 0.09 | -0.08 | 0.76 |
| **SLSTR vs VIIRS** | 0.07 | -0.06 | 0.81 |
| **VIIRS vs MODIS** | 0.04 | -0.018 | 0.87 |
| **SNPP/VIIRS vs TERRA/MODIS** | 0.04 | -0.01 | 0.81 |
| **SNPP/VIIRS vs AQUA/MODIS** | 0.04 | -0.001 | 0.83 |
| **NOAA20/VIIRS vs TERRA/MODIS** | 0.05 | -0.03 | 0.85 |
| **NOAA20/VIIRS vs AQUA/MODIS** | 0.04 | -0.02 | 0.87 |
| **PMAp-A vs TERRA/MODIS** | 0.08 | -0.001 | 0.55 |
| **PMAp-A vs AQUA/MODIS** | 0.08 | 0.007 | 0.55 |
| **PMAp-B vs TERRA/MODIS** | 0.06 | 0.03 | 0.67 |
| **PMAp-B vs AQUA/MODIS** | 0.07 | 0.04 | 0.66 |
| **SNPP/VIIRS vs NOAA20/VIIRS** | 0.03 | 0.02 | 0.91 |
| **AQUA/MODIS vs TERRA/MODIS** | 0.04 | -0.009 | 0.84 |
| **PMAp-A vs PMAp-B** | 0.09 | 0.03 | 0.54 |
| **S3B/SLSTR vs S3A/SLSTR** | 0.04 | 0.02 | 0.86 |






**Table 5: Quantification of AOD differences between instruments over the ocean regional domains (defined in Table 2) for the DJF period. Each cell gives the MD followed by the RMSD.**

|  | MA | NA | NP | SP | SIO | AI |
|---|---|---|---|---|---|---|
| **SLSTR vs MODIS** | **–0.08,0.1** | **–0.05,0.06** | **–0.05,0.05** | **–0.09,0.1** | **–0.09,0.1** | **–0.09,0.1** |
| **VIIRS vs MODIS** | **–0.02,0.06** | **–0.02,0.03** | **–0.02,0.03** | **0.002,0.04** | **–0.02,0.04** | **–0.02,0.04** |
| **PMAp vs MODIS** | **–0.01,0.09** | **0.005,0.2** | **–0.01,0.05** | **0.01,0.08** | **0.01,0.07** | **0.009,0.06** |
| **SLSTR vs VIIRS** | **–0.05,0.09** | **–0.03,0.05** | **–0.03,0.04** | **–0.09,0.1** | **–0.07,0.07** | **–0.07,0.08** |






### 4.1.2 First guess departure

Figure 5 and 6 present the global maps of the mean and SD of the first guess departure and Figures 7 and 8 display the
associated latitude cross sections by distinguishing the instruments on distinct platforms.

All satellite retrievals show negative global mean of first guess departure except PMAp. They exhibit consistent positive
departure over the dust and smoke plume off the West African coast (Fig. 5) which explains the increase in first guess departure
around the Equator shown by all the products in Fig. 7. VIIRS frequently shows more pronounced first guess departure than
MODIS over the remote oceans. SLSTR shows the largest negative first guess departure with values much lower than the
range spanned by the rest of the products (Fig. 7). The magnitude of SLSTR first guess departures increases in the South ocean
between ~35°S and ~50°S and in the mid-Pacific between 10 °N and 20 °N (Fig. 7). Consistently with what was shown for
AOD retrievals in Fig. 3, PMAp-B has larger and positive first guess departure than PMAp-A which has values close to
TERRA/MODIS. AQUA/MODIS has a negative offset compared to TERRA/MODIS which keeps first guess departure close
to zero. The first guess departure of SNPP/VIIRS is larger than that of NOAA20/VIIRS particularly between 15°S and 15°N
(Fig. 7) where it is close to that of AQUA/MODIS.

VIIRS shows sharp increases in first guess departure SD (Fig. 6 and 8) in the South Pacific (20°S to 50°S, 120°W to 160°W)
and off the Australian East coast which both are related to the Australian fire smoke. This is partially shown by SLSTR but
not by MODIS and PMAp (Fig. 8). While SLSTR has the largest magnitude of the mean of first guess departure it generally
has the lowest SD of first guess departure which indicates lower random differences with the model compared to the rest of
the products. PMAp frequently shows the largest SD of first guess departure, particularly in the Northern hemisphere, which
is related to the nosier patterns of PMAp retrieval compared to the rest of the products.  Fig.8 indicates good agreement of SD
of first guess departure between instruments on board distinct platforms except for PMAp-B which has larger values than
PMAp-A. Similar results are found for the MAM period (Figs. D.3, D.4, D.5 and D.6).

Figure 9 shows the statistics of the first guess departure for each product computed for distinct ranges of satellite AOD for the
DJF period (similar results are obtained for the MAM period). For AOD smaller than 0.2, SLSTR has a negative first guess
departure of about -0.05 while VIIRS and MODIS first guess departure values fall between -0.015 and 0. For AOD larger than
0.2, VIIRS, MODIS and PMAp show positive first guess departures. The mean and the spread of the first guess departure of
PMAp and VIIRS increase with AOD. MODIS shows lower values and smaller spread of first guess departure which is
expected because the model first guess is influenced by the assimilation of MODIS from the previous analysis cycles. SLSTR
has negative first guess departures up to AOD=0.4. The spread of SLSTR first guess is comparable to that of VIIRS.





**Figure 5: Global maps of the mean of first guess departure from TERRA&AQUA/MODIS, NOAA20&SNPP/VIIRS, S3A&S3B/SLSTR, Metop-A&B/PMAp, for the DJF (2019-2020) period, over ocean.**




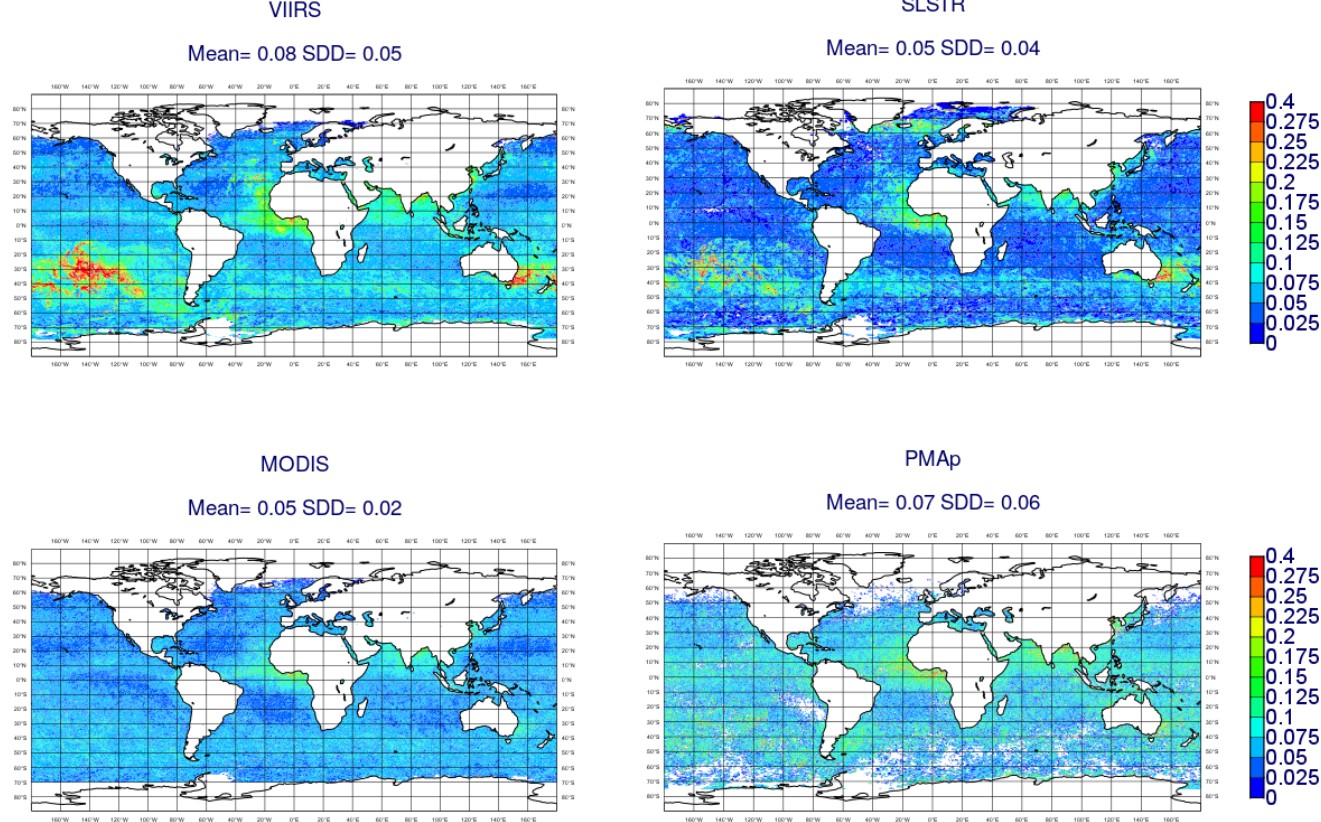

**Figure 6: Global maps of the standard deviation of first guess departure from TERRA&AQUA/MODIS, NOAA20&SNPP/VIIRS, S3A&S3B/SLSTR, Metop-A&B/PMAp, for the DJF (2019-2020) period, over ocean.**







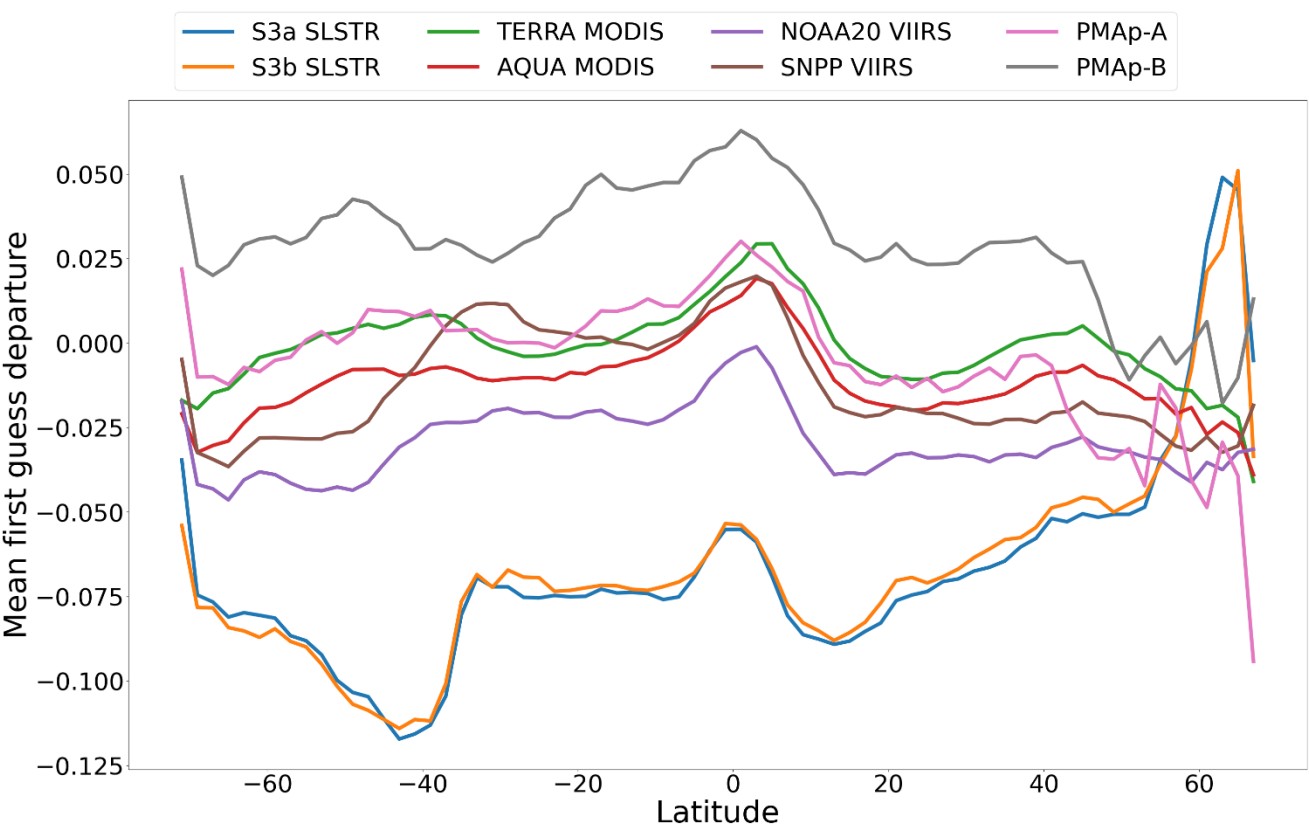


**Figure 7: Latitude cross-section of first guess departure mean, for the DJF (2019-2020) period, over ocean.**





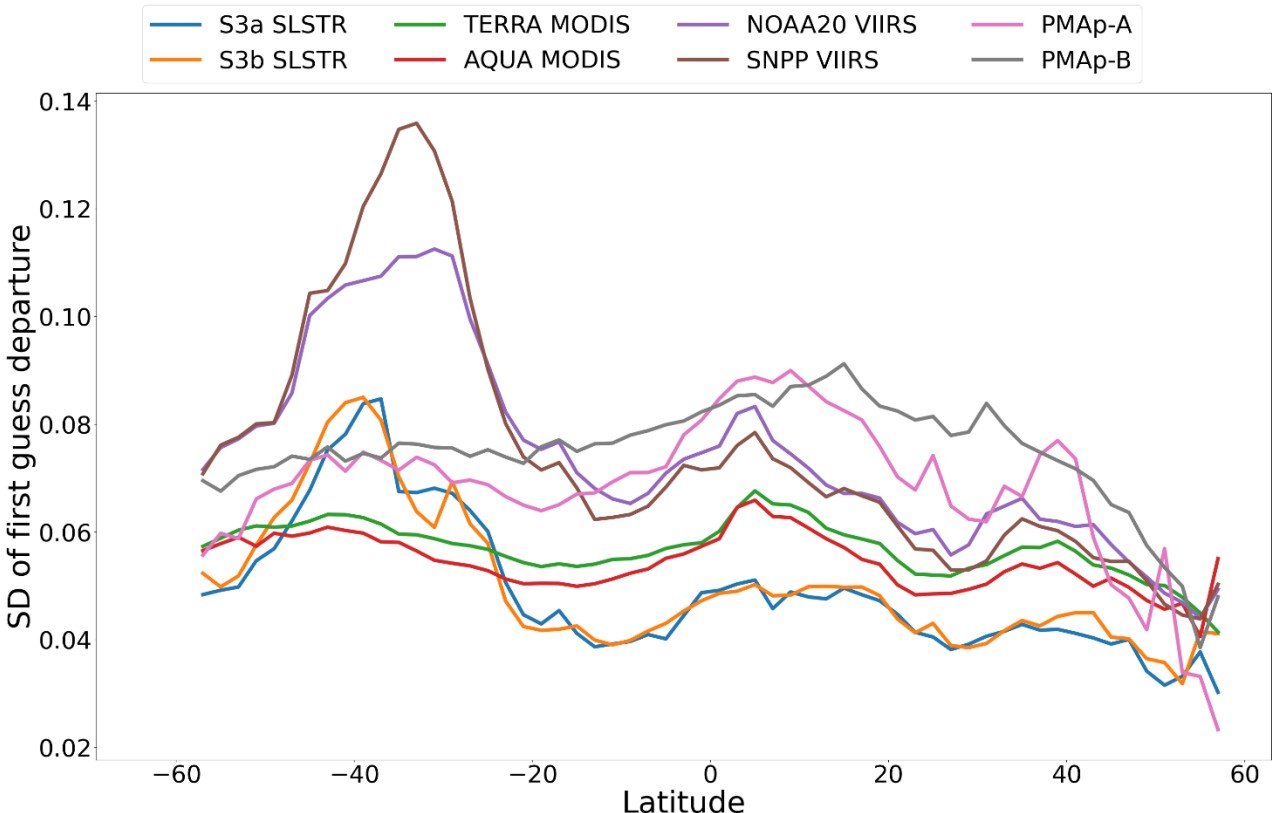


**Figure 8: Latitude cross-section of first guess departure standard deviation (SD), for the DJF (2019-2020) period, over ocean.**






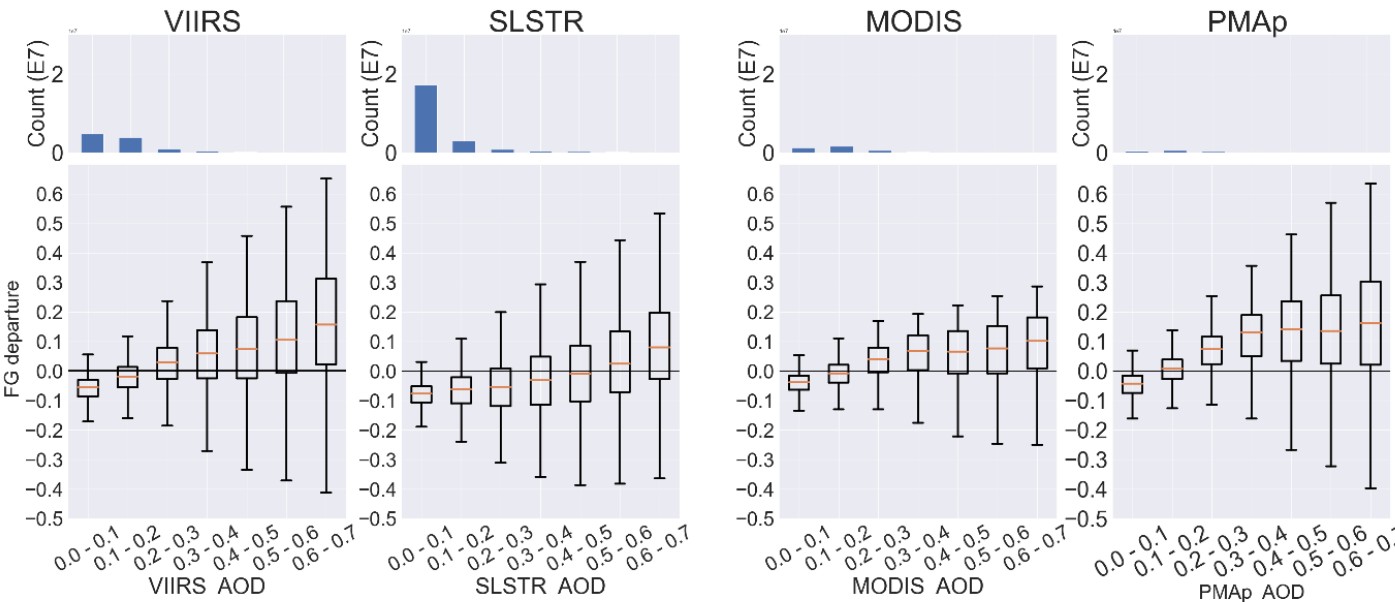

**445** **Figure 9: Global statistics of first guess departure (FG) for distinct range of AOD, for the DJF (2019-2020) period, over ocean**



## 4.2 Evaluation over land

### 4.2.1 Satellite observations

The magnitude of the AOD global mean of VIIRS and MODIS is larger over land than ocean and it increases from the DJF to the MAM period. Figure 10 and 11 show the overall good agreement between MODIS and VIIRS over land. VIIRS exhibits larger spatial coverage and smoother spatial variations over North latitudes, Central Africa and South America. Conversely to ocean, VIIRS AOD has a larger global mean than MODIS (difference of 0.03). VIIRS is higher over biomass burning regions (e.g. South-West coast of Australia for the DJF period, Central Africa and South America) and over dust source regions (e.g. Taklamakan desert for the MAM period, the Bodele depression for the DJF and MAM periods, the Sahel for the MAM period and Central Australia for the DJF period, Central America for the MAM period). VIIRS show smaller AOD than MODIS over the polluted hot spots in East China.

Figure 12 shows larger differences in global AOD PDF shapes between MODIS and VIIRS over land compared to ocean. MODIS has a wider and quasi bi-modal distribution while VIIRS exhibits a positively skewed distribution. All VIIRS regional distributions are positively skewed toward larger AOD values than MODIS except over Africa where MODIS and VIIRS show similar PDF (Fig B.2). The latitude transects across land surfaces (Fig. 13 for the DJF period and Fig. C.2 for the MAM period) indicate higher VIIRS AOD than MODIS AOD in the Southern Hemisphere where the differences are larger than in the Northern Hemisphere. The AOD peak around 40°S related to the Australian fires in Fig. 13, is more pronounced for VIIRS than for MODIS. Table 7 indicates smaller differences between VIIRS and MODIS over Europe, Africa and North-America while the largest MD and RMSD are obtained over Australia and South-America. SNPP/VIIRS and NOAA20/VIIRS show better agreement over land than over ocean except in the 20° to 30° North and 20° to 30° South latitude bands where NOAA20/VIIRS is larger and lower, respectively, than SNPP/VIIRS for the DJF period (Fig. 13). AQUA/MODIS and TERRA/MODIS AOD are in close agreement in the northern hemisphere up to 30°N above which AQUA/MODIS AOD drops below TERRA/MODIS for the DJF period (This is not observed for the MAM period). In the Southern Hemisphere, AQUA/MODIS is systematically higher than TERRA/MODIS while the opposite was observed over ocean. Overall the departure between retrievals from the same instrument on board different platforms is less important over land than over ocean (Table 6 compared to Table 4).



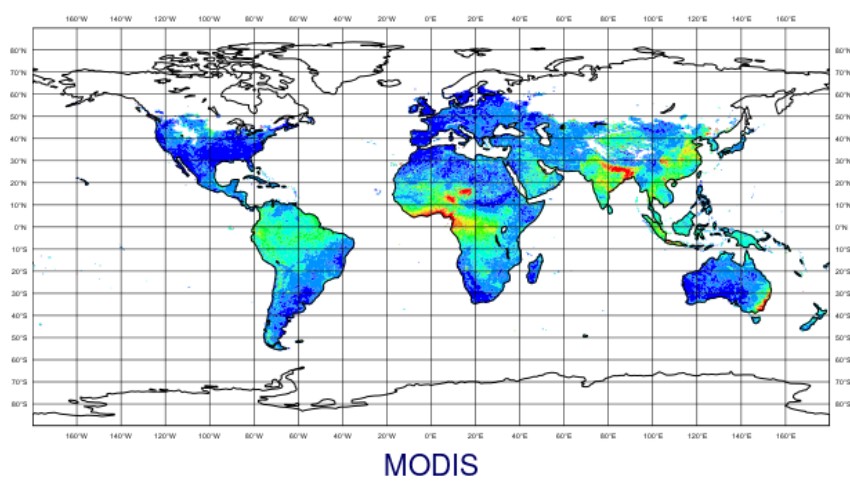

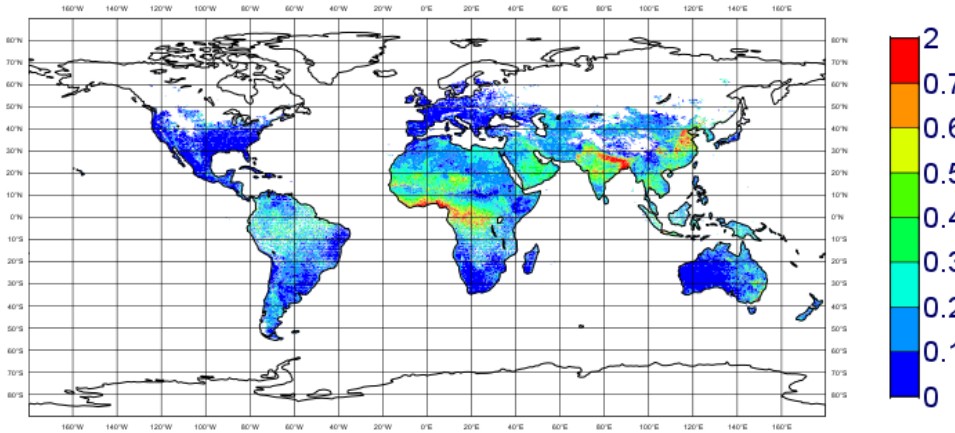


**Figure 10: Global maps of temporal mean AOD from TERRA&AQUA/MODIS, NOAA20&SNPP/VIIRS, for the DJF (2019-2020) period, over land.**





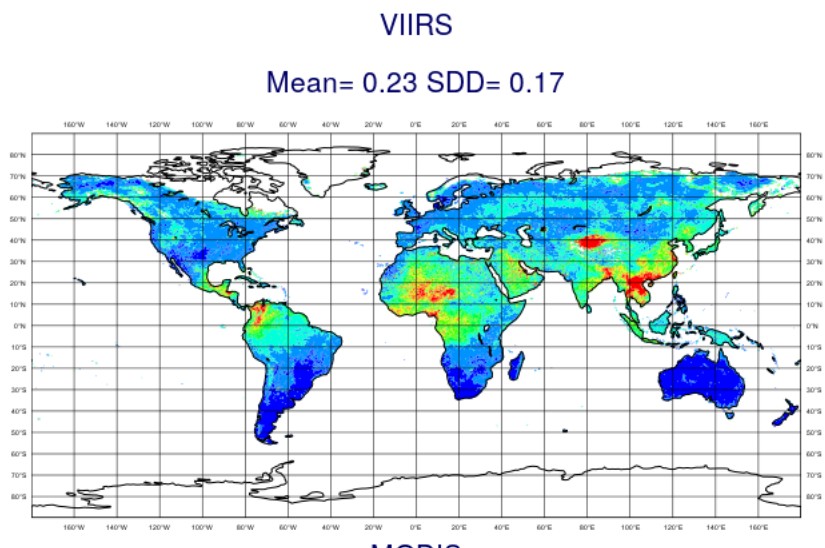

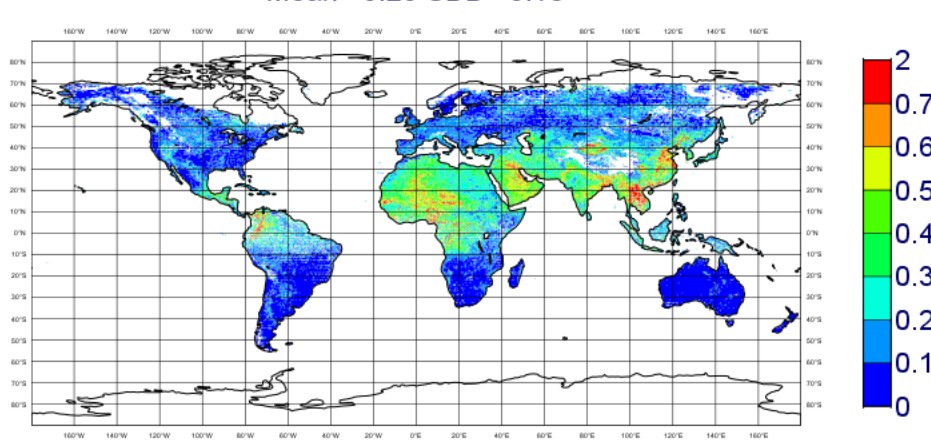

**Figure 11: Global maps of temporal mean AOD from TERRA&AQUA/MODIS, NOAA20&SNPP/VIIRS, for the MAM (2020) period, over land.**








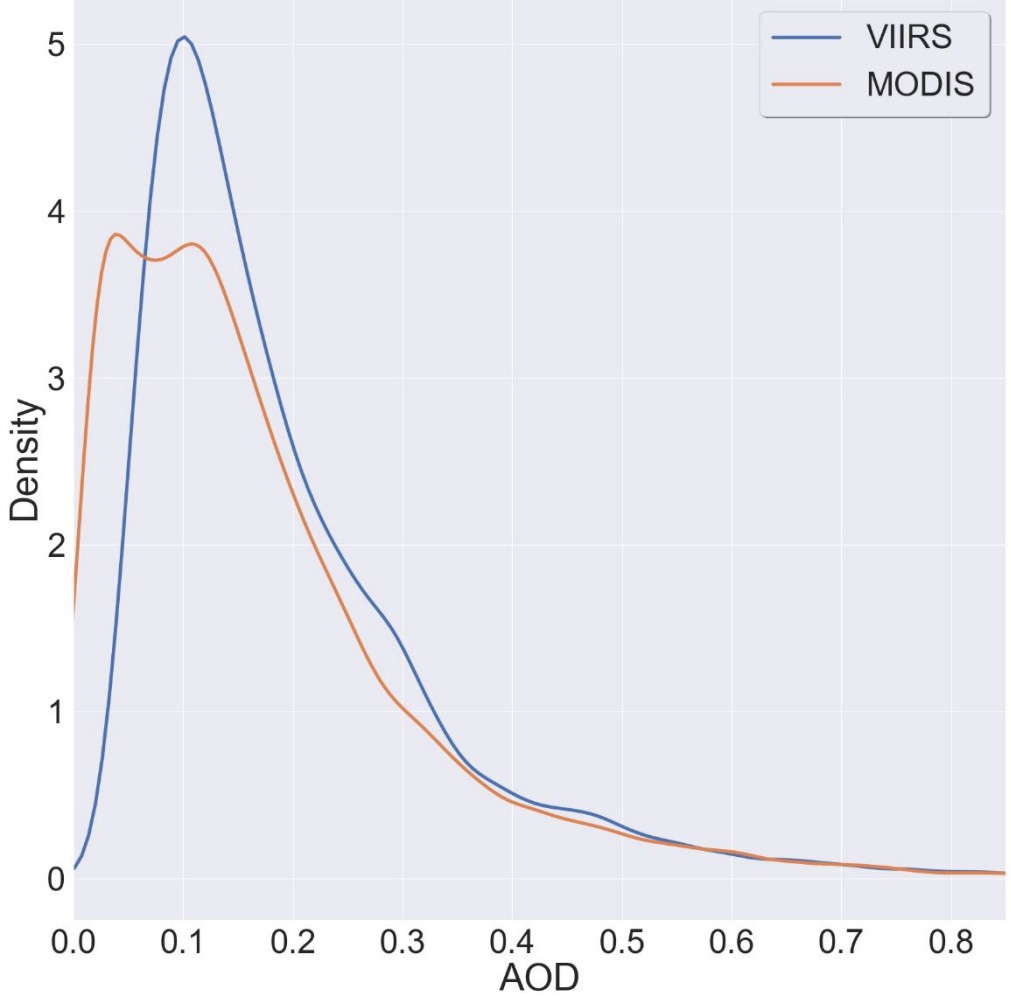

**Figure 12: Global satellite AOD distribution over land (DJF period).**






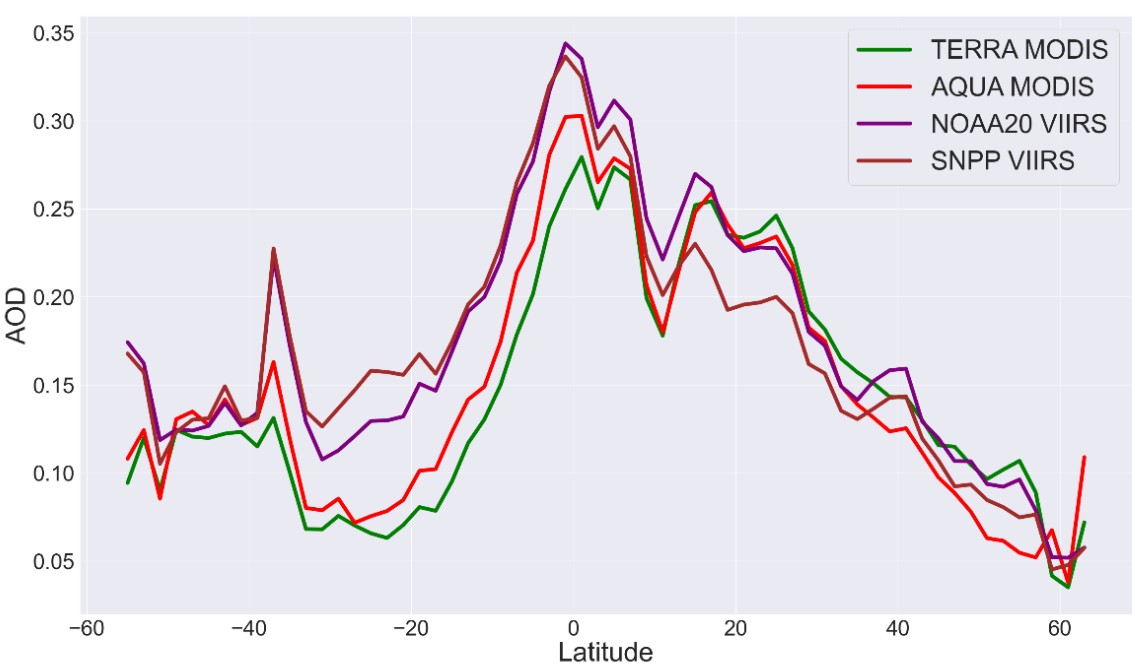

**Figure 13: Latitude cross-section of temporal mean satellite AOD, for the DJF (2019-2020) period over land.**





**Table 6: Quantification of global AOD differences between VIIRS vs MODIS over land for the DJF and MAM periods**

|  | RMSD | MD | r |
|---|---|---|---|
| **VIIRS vs MODIS - DFJ period** | 0.10 | 0.025 | 0.76 |
| **VIIRS vs MODIS - MAM period** | 0.12 | 0.032 | 0.76 |





**Table 7: Quantification of AOD differences between instruments over the land regional domains (defined in Table 3) for the DJF period. Each cell gives the MD followed by the RMSD.**

|  | AF | AS | NA | SA | EU | AU |
|---|---|---|---|---|---|---|
| **VIIRS vs MODIS** | 0.0007,0.1 | 0.04,0.1 | 0.04,0.06 | 0.05,0.1 | 0.02,0.07 | 0.04,0.1 |
| **SNPP/VIIRS vs TERRA/MODIS** | -0.01,0.1 | 0.008,0.12 | 0.0016,0.058 | 0.081,0.12 | -0.02,0.07 | 0.09,0.1 |
| **SNPP/VIIRS vs AQUA/MODIS** | -0.023,0.11 | 0.022,0.11 | 0.003,0.06 | 0.060,0.10 | 0.008,0.06 | 0.06,0.13 |
| **NOAA20/VIIRS vs TERRA/MODIS** | 0.01,0.11 | 0.02,0.11 | 0.006,0.06 | 0.07,0.11 | -0.01, 0.07 | 0.067, 0.13 |
| **NOAA20/VIIRS vs AQUA/MODIS** | 0.001, 0.11 | 0.034,0.12 | 0.036, 0.06 | 0.05,0.10 | 0.020, 0.07 | 0.04,0.12 |




### 4.2.2 First guess departure

Figure 14 indicates more frequent positive and negative first guess departures for VIIRS and MODIS, respectively, for the DJF period. This holds for the MAM period (Fig D.7). VIIRS shows more pronounced positive first guess departure over South America, Central and South Africa, South East Australia (DJF only), West Australia, North America (particularly for

the MAM period) and the Taklamakan region. Both MODIS and VIIRS consistently show negative departure in Central Africa for the DJF period (Fig. 14) and in India for the MAM period (Fig D.7) which corresponds to the decrease in first guess departure between 10°N to 15°N shown by the latitude transects in Fig. 16 and D.9. Figure 16 indicates larger differences in first guess departure between instruments and platforms in the Southern hemisphere compared to the Northern hemisphere. In the Southern hemisphere, MODIS and VIIRS frequently show negative and positive, respectively, first guess departure. The

first guess departure magnitude of TERRA and SNPP are larger than that of AQUA and NOAA20, respectively. In the Northern hemisphere, both VIIRS and MODIS have negative departure up to ~40°N. Above 40°N, VIIRS and TERRA/MODIS consistently show slightly positive first guess departure while AQUA/MODIS keeps negative values.

The SD of first guess departure (Fig. 15 and 17) is overall larger for VIIRS than MODIS (VIIRS has a global average twice that of MODIS). Figure 15 and D.8 highlight the large differences between VIIRS and the model AOD over dust sources

regions (Africa, Middle East, Asia, Australia), biomass burning regions in Africa and polluted regions in India and China. While TERRA and AQUA have similar SD of first guess departure, NOAA20 shows larger values than SNPP in the Northern Hemisphere (Fig. 17 and Fig. D10).

Figure 18 indicates that MODIS and VIIRS have both similar first guess departure statistics for AOD less than 2. For AOD larger than 2, the mean and the variance of the VIIRS first guess departure increases with AOD while the MODIS first guess

departure is steady and less variable due to its assimilation.



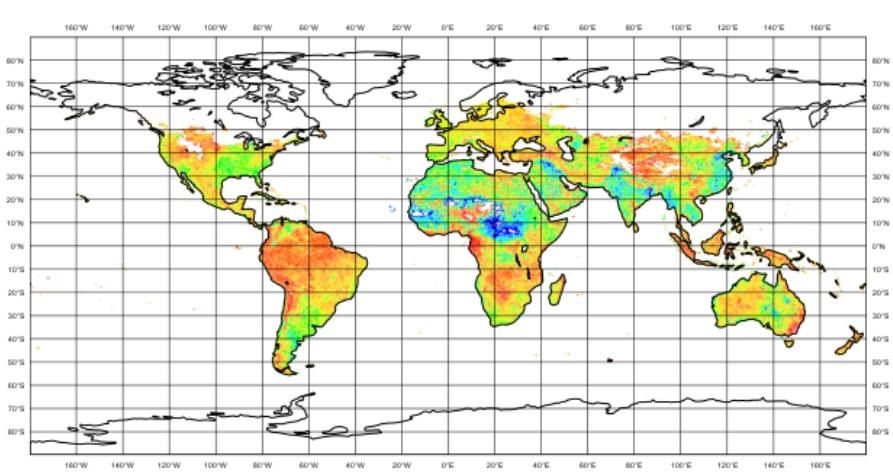

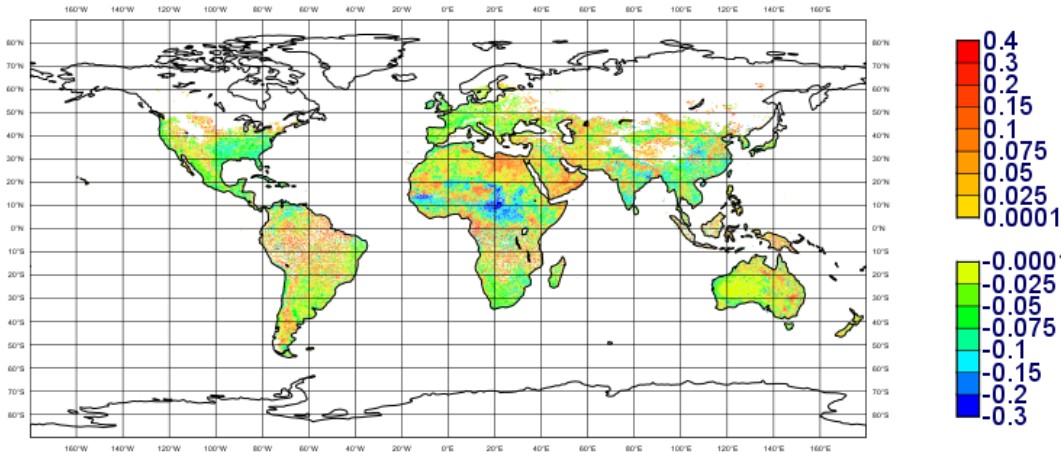


**Figure 14: Global maps of the mean of first guess departure from TERRA&AQUA/MODIS, NOAA20&SNPP/VIIRS, for the DJF (2019-2020) period, over land.**






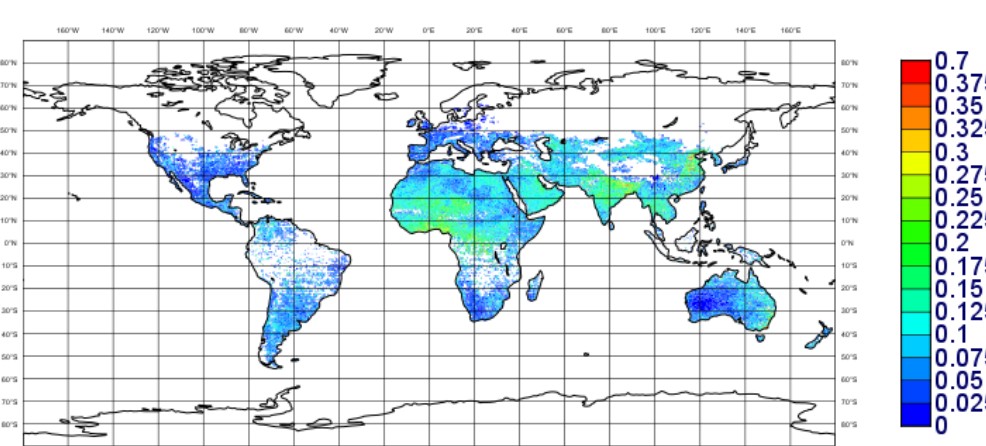

**Figure 15: Global maps of the standard deviation of first guess departure from TERRA&AQUA/MODIS, NOAA20&SNPP/VIIRS, for the DJF (2019-2020) period, over land.**







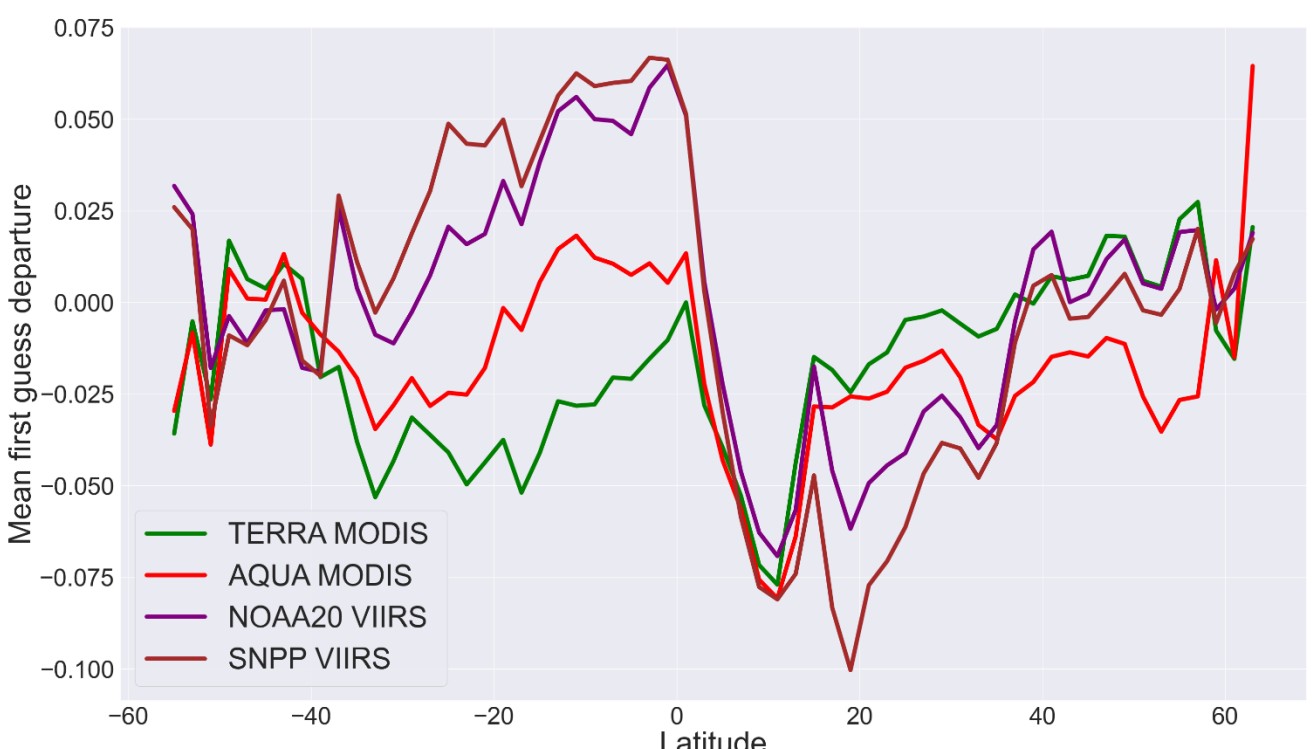


**Figure 16: Latitude cross-section of first guess departure mean, for the DJF (2019-2020) period over land.**







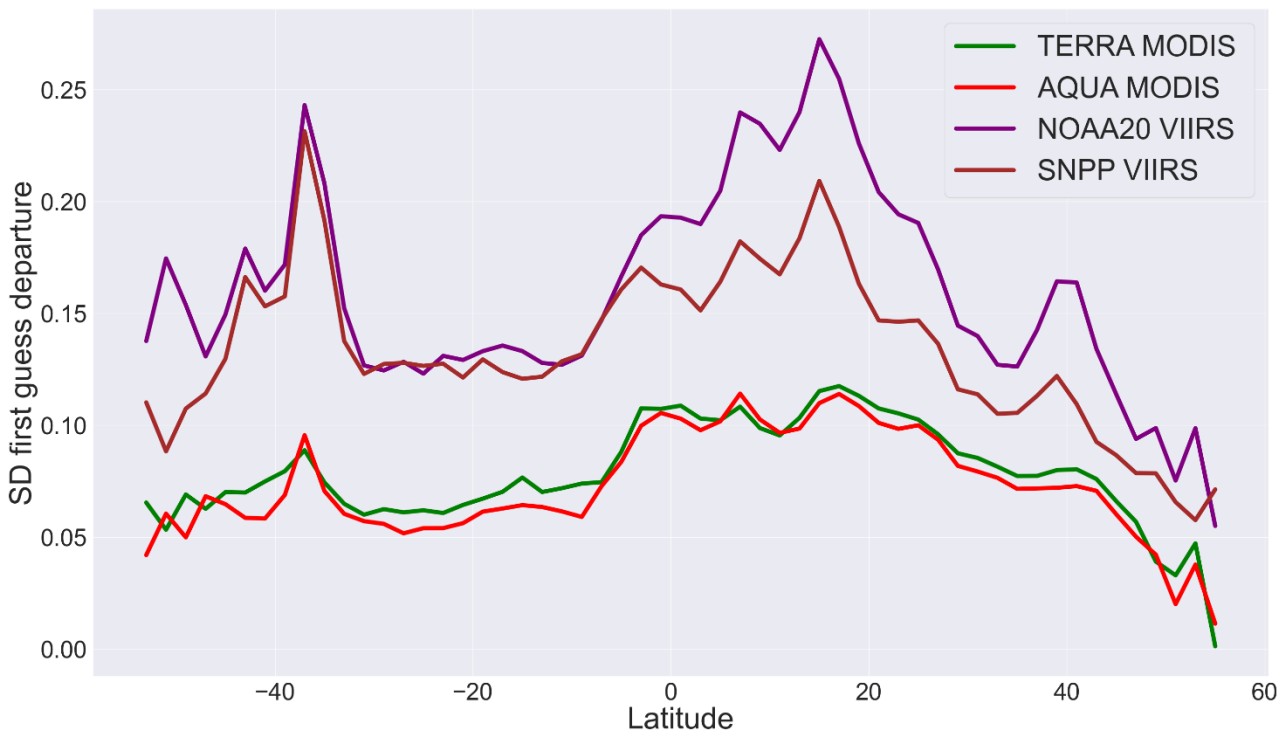


**Figure 17: Latitude cross-section of first guess departure standard deviation (SD), for the DJF (2019-2020) period over land.**





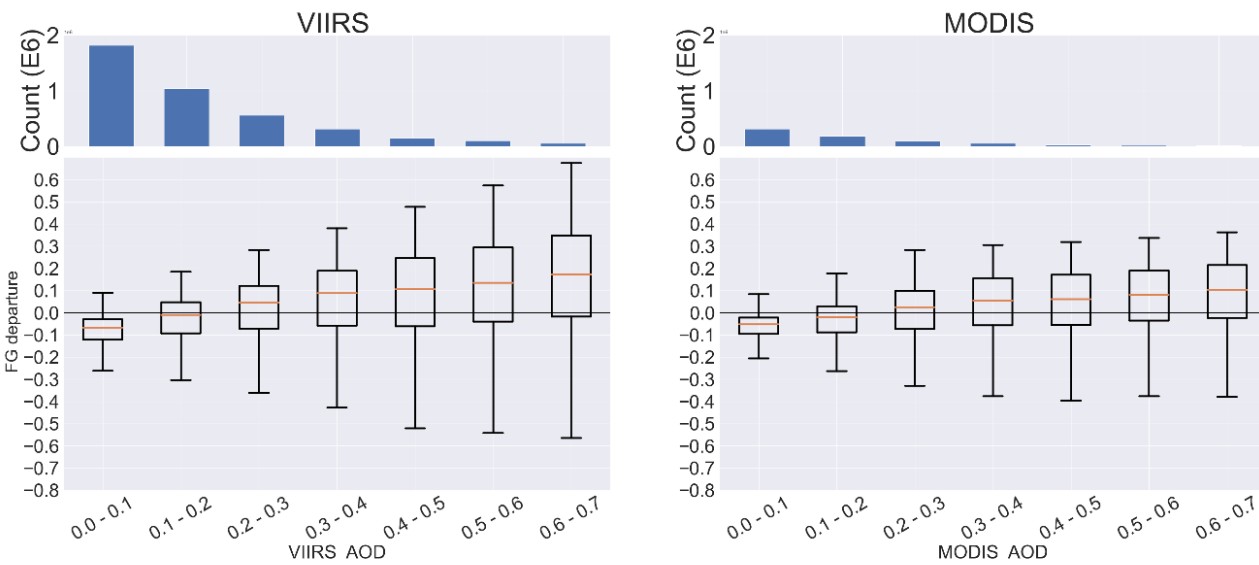


**Figure 18: Global statistics of first guess (FG) departure for distinct range of AOD (DJF period) over land.**






## 5 Discussion

The sources of differences between the satellite AOD products monitored within the CAMS data assimilation system are discussed here.

### 5.1 Cloud detection

Cloud contamination has been identified as an important source of uncertainties in aerosol retrieval (Kaufman et al., 2005; Li et al, 2009; Sogacheva et al., 2017; Schutgens et al., 2020; Garay et al., 2020). Commission errors are frequent between cirrus and dusts (Lee et al., 2013) or in case of heavy smoke (Wong and Li, 2002). Zhao et al., 2013 have reported changes in monthly mean MODIS AOD up to 0.04 due to cloud contamination.

While the gridding of the satellite products within the CAMS data assimilation system should minimize the impacts of
differences in spatial resolution between products, the differences in cloud filtering can lead to substantial differences in AOD spatial representativity between products at the model grid spatial resolution. Differences in spatial representativity can generate differences in AOD which can be larger than the differences between collocated level-2 retrievals (Virtanen et al., 2018; Schugens et al., 2016,2019). The much lower AOD shown by SLSTR over the ocean background is probably related to the stringent cloud filtering applied to the SLSTR L1B radiances. The SLSTR product relies on the native L1B cloud mask
which was originally designed for Sea Surface Temperature retrieval and was proven to be too conservative for aerosol retrieval (SLSTR PVR, 2021). The SLSTR cloud mask frequently removes medium values of L1B radiances and thus medium AOD values over the ocean (SLSTR PVR 2021), which leads to substantially reduced level-3 AOD values at the model grid resolution and explains the fragmented aspect of the aerosol plumes displayed by the level-3 SLSTR product. The negative departures between SLSTR and the model are more pronounced in the Southern and Northern hemispheres for the DJF and
MAM periods, respectively, which indicates a possible seasonality in the differences in spatial representativity due to cloud filtering. The magnitudes of the differences between the level-3 SLSTR and MODIS AOD over ocean reported in this study are mainly dominated by the differences in spatial and temporal representativity at the model spatial resolution due to cloud overscreening in the SLSTR product, which explains their larger values (MD of -0.08 and RMSDD of -0.09) than the ones indicated in the SLSTR validation report for level 2 retrievals (SLSTR PVR, MD= -0.03 and RMSD=0.05). A new cloud mask
tuned for aerosol retrieval over ocean is under development at EUMETSAT, which should improve the consistency of level-3 AOD with the rest of the products at low AOD. While SLSTR shows under-representativeness issues, cloud residuals in VIIRS and MODIS can increase locally their 3-month average AOD value which also contributes to the differences between products. The differences between VIIRS and MODIS in the North Atlantic, where high cloud cover is frequent during the DJF period, can also be related to differences in cloud detection. The higher AOD values from TERRA than AQUA over land at a latitude
higher than 30°N can be due to the diurnal variation of cloud contamination (Painemal et al., 2015) which is generally more frequent at the early morning overpass of TERRA during the DJF period. The use of a heavy aerosol detection test in the VIIRS algorithm reduces the commission errors between cloud and optically thick aerosols, which partly explains why VIIRS resolves the smoke plume in the Pacific where MODIS and PMAp show nosier spatial patterns due to cloud residuals. Finally,



the higher spatial resolution of VIIRS and its reduced pixel deformation at the edge of the swath should improve cloud
detection. This was demonstrated for the MAIAC MODIS product which has a reduced cloud detection commission error
compared to the MODIS standard product (Lyapustin et al., 2012, Jethva et al. 2019).

**5.2 Instrument geometry**

Geometry is a key factor to understand the uncertainty in AOD retrieval since it influences the range of scattering angles
sampled by the instrument and thus the degree of information content available for the retrieval (Foughie et al., 2020, more
details in Appendix A.1).

Since the uncertainties in AOD retrieval vary with view angle and the length of the atmospheric path, the retrieval artefacts at
the edge of the swath are expected to be larger for Metop-B, which has a double swath compared to Metop-A. Besides, the
differences in swath can generate distinct ranges of scattering angles sampled by the instrument, which contributes to the
differences in AOD retrieval between Metop-A and -B.

Despite the similarity between VIIRS and MODIS instruments, VIIRS has a smaller pixel deformation at the edge of the swath
which should limit geometry-induced biases compared to MODIS. Besides, the finer spatial resolution, at which VIIRS
retrieval is performed, and its larger swath, imply more frequent retrievals compared to MODIS (Sayer et al., 2018) and explain
the larger spatial coverage of VIIRS over Northern latitudes, Central Africa and South America. The VIIRS higher spatial
resolution allows to resolve finer spatial details such as the Australian fire smoke transport in the Pacific which is not detected
by MODIS and PMAp.

While one could expect a North/South structure bias in SLSTR retrievals due to more frequent unfavourable geometries
(backscattering region) of the oblique view in the Northern Hemisphere (Foughie et al., 2020), this is not shown in our results
over ocean. But the impact could be stronger over anisotropic land surfaces. Other factors may also influence the information
content of the SLSTR dual-view which varies not only along the swath in the North-South direction but also across the swath
in West-East direction and with seasons.

**5.3 Measurement information content**

VIIRS and MODIS are two imaging radiometers characterized by similar spectral information content, which can explain the
overall better agreement between VIIRS and MODIS level-3 AOD compared to the rest of the products. This is consistent with
the conclusions from Sayer et al., 2018 who showed that MODIS and VIIRS products capture similar temporal and spatial
variations and have similar level of uncertainty evaluated against AERONET.  However, slight differences in spectral bands
(e.g. blue bands) and the associated spectral response functions can play a role in the differences between MODIS and VIIRS
particularly at low AOD.
The GOME-2 instrument, which provides the main measurement to retrieve PMAp AOD, relies on a very different
measurement technique compared to VIIRS and MODIS, with spectral information in the UV-VIS channel, which can explain
part of the differences between PMAp and VIIRS or MODIS retrievals.



The dual-view capability of SLSTR offers a potentially larger number of independent, cloud-free and glint-free observations over ocean. However, the lower spectral information content of SLSTR (e.g. lack of a blue band) compared to VIIRS and

MODIS can be a limitation to accurately separate out the surface contribution such as the impact of in-water scattering and absorption from plankton pigments that can explain some of the differences observed in the South ocean.

**5.4 Radiometric calibration**

AOD retrieval requires high radiometric accuracy of the input reflectance and consistency across bands and views. Small

differences in sensor calibration and spectral response functions can have a large impact on AOD retrieval (Kaufman et al, 1997), particularly for the ocean background associated with low AOD (Sayer et al., 2018). The impact is larger for bright than dark surfaces since the error scales with the magnitude of the surface reflectance (Zhang et al.,2016). Upstream radiometric calibration uncertainties can explain a large part of the differences between retrievals from similar algorithm and instrument but from distinct platforms (Jourdan et al., 2007., Levy et al., 2013., Sayer et al., 2017). The positive offset of

TERRA over ocean compared to AQUA (Fig. 4), which has been acknowledged by various studies (Levy et al., 2018, Sogacheva et al., 2019), is partly related to a larger radiometric calibration degradation for TERRA than AQUA which is not corrected in the DT retrieval algorithm (Sayer et al., 2018). Besides, reflectances in the solar reflective bands of SNPP/VIIRS have also been found systematically higher than those of NOAA20/VIIRS which has a more consistent inter-channel calibration and a steadier calibration in time compared to SNPP/VIIRS (Sirish et al., 2020). Over land, where the surface

reflectance and AOD are retrieved simultaneously from observed VIIRS TOA reflectance, the SNPP/VIIRS vs. NOAA20/VIIRS reflectance difference does not necessarily lead to a corresponding AOD difference. In contrast, over ocean, the surface reflectance is calculated from a model that is independent of instrument calibration, and so a positive bias in the SNPP/VIIRS TOA reflectance directly translates into a higher AOD retrieval compared to NOAA20/VIIRS. The positive offset between PMAp-B and PMAp-A is also related to differences in GOME-2 radiometric performances (e.g. dark current,

straylight, polarisation) between METOP-A and METOP-B. A correction has been implemented in the new release 2.2.4 of PMAp (Grzegorsk et al., 2021) that will be assessed in a future work. The good agreement between S3A and S3B retrievals is explained by a radiometric alignment which has been implemented in the SLSTR L1B processing after the tandem campaign between S3A and S3B in June-October 2018. Radiometric calibration residuals can also explain part of the bias of SLSTR product at very low AOD (SLSTR PVR, 2021).


**5.5 Surface reflectance parametrization**

Over the ocean background, AOD retrieval at low AOD (less than 0.2) is very sensitive to small errors in the surface reflectance which are frequently due to uncertainties in near-surface wind speed parametrization (Sayer et al., 2018). This can explain the large diversity in AOD values observed over the South ocean where large near surface wind speed are frequently

underestimated by meteorological forecast data sets (Bentamy et al., 2021). Besides, differences in wind speed between meteorological data sets (NCEP for VIIRS and MODIS versus ECMWF for SLSTR and PMAp) can also play a role in the





differences between retrievals. Finally, the way the wind speed is accounted for in the LUT can influence the retrieval. For example, MODIS exploits 4 fixed wind speed values (2, 6, 10, 14 m s$^{-1}$) to generate the surface reflectance which may be too coarse to accurately represent the impact of wind speed variability. Similarly, inaccuracies in the wind speed and wind direction

influence the estimation of the glint and white cap components of the ocean BRDF model which have been identified as possible sources of the negative bias of the SLSTR product at low AOD (SLSTR PVR, 2021).

Over land, surface reflectance models may fail to represent the angular variations of surface reflectance over highly complex land terrain or the seasonal and interannual variability of the surface reflectance *(*Kokanovsky et al., 2007, Liu et al., 2014; Huang et al., 2016; Tao et al., 2017). The underestimation of AOD retrieved from the MODIS DB algorithm over desert can

be partly related to inaccurate representation of dust regional variability in the surface reflectance database used in the MODIS DB retrieval (Hsu et al., 2019).

## 5.6 Aerosol models

Inconsistencies in aerosol properties between algorithms can generate large differences in AOD retrieval (Kokanovsky et al.,

2007; Levy et al., 2013). Most retrieval algorithms rely on a limited number of broad classes of aerosol models, which may not be sufficient to represent the spatial and temporal variability of actual aerosol properties such as the large variability in particle shape, size and mineralogy composition of dusts, the large variations in aerosol optical properties over polluted regions due to the impact of transport, aging and secondary organic aerosol processes and the variability in smoke properties related to type of fuel and surface moisture conditions (Shi et al., 2011, Ichoku et al., 2003, Sayer et al., 2013, Huang et al., 2016).

While spherical assumption is a good approximation for sulphate and carbonaceous aerosols (Martins et al., 1996), it can lead to geometry- and spectral- dependent biases in AOD for dusts (Mishchenko et al., 1995, Torres et al., 1998., Zhang and Reid, 2005, Levy et al., 2013; Lee et al., 2017 , Sayer et al., 2018, Zhou et al., 2020).

VIIRS and MODIS exploit distinct aerosol models which can explain part of the larger VIIRS AOD values over biomass burning and desert regions. Tao et al., 2017 showed that the dust scattering properties are overestimated in the MODIS DB

algorithm which results in a negative AOD bias over desert regions. Besides, the spherical dust model used in the DT over ocean was shown to introduce a positive bias in case of high dust load (Zhou et al., 2020) which could explain the larger MODIS AOD than VIIRS over the dust outbreak in the mid-Atlantic (5°N-10°N, Figure 4). This can also play a role in the differences between MODIS and SLSTR (SLSTR validation report) but it is probably of second order compared to the differences in spatial representativity. Finally, our results indicate differences between MODIS and VIIRS over the polluted

hot spots in China that can be related to differences in the fine mode aerosol models used in both retrieval algorithms.

## 5.7 Spatial and temporal differences between products

Product diversity differs over land and ocean as illustrated between VIIRS and MODIS which have opposite MD over ocean and land. For land, the source of differences between products at large AOD is primarily related to the aerosol models and the

representation of the surface reflectance anisotropy while for the ocean background the sources of differences at low AOD



mainly arise from small differences in the calculated surface reflectance mainly due to cloud contamination, calibration uncertainties and inaccurate wind speed parametrization (Zhang and Reid, 2005, Smirnov et al., 2009; Sayer et al., 2018).

Over land, retrievals are frequently more uncertain and more diverse over bright (e.g. bare, desert), complex (e.g. urban, mountains) and elevated terrains, where the strong surface anisotropy requires a higher degree of information content to retrieve AOD (de Leeuw et al, 2018, Schutgens et al., 2021). Our results show lower MODIS values than VIIRS over the Taklamakan desert, the Bodele depression, the Sahel and Central Australia which represent major global dust sources. This confirms results from Tao et al., 2017 who reported a negative bias for MODIS DB over South East Asia desert regions. Besides, Sayer et al., 2018 showed larger VIIRS DB AOD than MODIS DB over dust regions (Sahara, Arabian Peninsula, and Taklamakan desert) that the authors relate to the improved aerosol and surface reflectance models used in the VIIRS DB algorithm.

Over ocean, product diversity increases in the South ocean where PMAp and MODIS exhibit noisy spatial patterns and SLSTR has larger departure with the model. A systematic positive AOD anomaly, referred as the Enhanced Southern Oceans Anomaly (ENSOA), which have been reported for various satellite AOD products (e.g MODIS, MISR) over mid-to-high latitude Southern Oceans (45° S to 65° S), is likely due to unfiltered stratocumulus and low broken cumulus clouds, inaccuracy in ocean surface albedo assumptions, high wind speed, inaccurate aerosol models, and floating ice (Zhang and Reid, 2005., Shi et al., 2011., Toth et al., 2013).

Finally, our results indicate slightly larger diversity between AOD products for the MAM period compared to the DJF period over land which is related to seasonality in dust and biomass burning events. However, while both DJF and MAM periods encompass a range of aerosol events which are representative of global aerosol variability, AOD products should be monitored over a longer period in further works to better resolve the seasonal and interannual variability of aerosols such as the North America and Siberia fires during the June-September period.

## 6 Conclusion

The objective of this work is to evaluate two new NRT satellite AOD products to prepare their assimilation in the CAMS data assimilation system, namely the Copernicus SLSTR AOD (C1) from Sentinel-3A&B over ocean and the NOAA EPS VIIRS AOD (v2r1) from S-NPP and NOAA-20 over both land and ocean. The diversity between MODIS (C6.1), PMAp (v2.1), VIIRS (v2r1) and SLSTR (C1) AOD products as well as their differences with the model (IFS CY47R1) first guess were assessed separately over land and ocean at the model grid resolution (level-3) using 3-month AOD average (December 2019 - February 2020 and March - May 2020). The outcomes of this work concern level-3 AOD from the perspective of their use in the CAMS data assimilation system, which may not directly apply to level-2 retrievals at their native spatial and temporal resolution.

Over ocean, VIIRS and MODIS show the best agreement among the investigated products. VIIRS is smaller than MODIS for the ocean background (global MD between VIIRS and MODIS is -0.02). VIIRS generally shows more pronounced negative



departure with the model than MODIS over the remote oceans. SLSTR has the lowest AOD (MD between SLSTR and VIIRS (MODIS) is -0.06 (-0.09)). PMAp shows the largest variability at global scale and the largest discrepancies between

instruments on board distinct platforms: PMAp-B has a large positive offset compared to the other products while PMAp-A is closer to TERRA/MODIS. Our results indicate a positive departure between TERRA/MODIS and AQUA/MODIS, and between SNPP/VIIRS and NOAA20/VIIRS. All the products capture the mid-Atlantic dust outbreak and the smoke plume from Central Africa. However, only VIIRS and SLSTR capture the smoke transport over the Pacific originated from the December 2019 - January 2020 Australian fires.

Over land, overall, VIIRS and MODIS show consistent global spatial distributions. The main differences appear in dust source (e.g. Taklamakan desert, Bodele depression, Sahel, Central Australia, Central America) and biomass burning (e.g. Australia, Central Africa, South America) regions where VIIRS AOD is frequently larger than MODIS and shows larger positive departure with the model. Better consistency between MODIS and VIIRS is observed in the Northern Hemisphere than in the Southern Hemisphere. VIIRS AOD is smaller than MODIS over the polluted hot spots from East China.

The main sources of diversity between retrievals over land at large AOD (e.g. dust and biomass burning regions) are mainly related to the differences in aerosol models (e.g. refractive index and particle size) and the representation of the surface reflectance anisotropy while for the ocean background, which is generally characterized by low aerosol burden, differences between retrievals mainly arise from uncertainties in cloud detection, radiometric calibration and surface reflectance model parametrization (e.g. wind particularly in the South ocean). While the magnitude of the mean deviation between the products

is smaller over ocean than over land, given the low AOD value of the ocean background, a slight difference in AOD between products will have a large impact on data assimilation. Besides, cloud filtering can generate large differences in spatial and temporal representativity between products at the model grid spatial resolution. The stringent cloud mask used in the SLSTR product explains the smaller level-3 AOD values of SLSTR compared to the rest of the products. The use of a heavy aerosol detection test helps to reduce cloud contamination commission errors as demonstrated by the detection of the smoke transport

over the Pacific by the VIIRS product. The consistency in cloud filtering between products should be properly evaluated and improved to minimize the differences in spatial representativity at the model grid spatial resolution. A compromise should be found between i) a strict enough cloud filtering to assimilate best quality retrievals and ii) enough spatial coverage to resolve the aerosol plumes and properly sample the global and regional AOD distribution. The similarities between VIIRS and MODIS radiometers and their associated retrieval algorithms explain their good agreement compared to PMAp and SLSTR which rely

on different measurement techniques and retrieval algorithms. The geometry characteristics of the instruments (swath, spatial resolution, view angles), which drive the range of scattering angles sampled by the instrument, is an additional source of differences between retrievals. The positive offset between PMAp-B and PMAp-A is partly related to the larger swath of Metop-B/GOME-2 which has larger retrieval artefacts at the edge of the swath. Similarly, more accurate retrieval is expected for VIIRS due to the reduced pixel deformation at the edge of the swath compared to MODIS. Besides, the finer spatial

resolution at which VIIRS retrieval is performed plays a role in the detection of the smoke transport over the Pacific where the PMAp and MODIS coarser products show nosier patterns. While the retrievals from the oblique view of SLSTR are expected





to be more accurate in the Southern Hemisphere, where forward scattering angles are more frequently sampled by the instrument, no North/South structure in the departure between SLSTR retrieval and the rest of the products was noticed in our level-3 evaluation over ocean. Finally, uncertainties in upstream radiometric calibration is a major source of differences
between retrievals from the same instrument but on-board distinct platforms as shown by the positive offset over ocean between TERRA/MODIS and AQUA/MODIS retrievals due to the non-corrected radiometric calibration degradation of TERRA/MODIS in the DT algorithm and between SNPP/VIIRS and NOAA20/VIIRS retrievals related to the positive bias in the solar reflective bands of SNPP/VIIRS.






**Appendices**

**Appendix A: Satellite AOD products**

**A.1 General characteristics**

The main difficulty for retrieving AOD from TOA reflectance measurements is to disentangle the respective contributions of

the aerosol and the surface reflectance. The measurement information content available to retrieve AOD is strongly constrained by the domain of the aerosol phase function which is sampled by the instrument (Fougnie et al., 2020). Retrieval conditions are generally more favourable in the forward domain, for which the amplitude of the aerosol signal is the largest, the sensitivity of the phase function to the aerosol models is low and the surface signal is weak. Retrieval is more complex in the backscattering region where the TOA reflectance is dominated by the surface signal and the retrieval is very sensitive to

uncertainties in the aerosol models. The range of scattering angles resolved by the instrument varies along track (North/South direction), across track (West/East direction) and with the season (Fougnie et al., 2020). Complexity increases over land bright surfaces, where geometrical scattering from individual surface elements with size larger than the wavelength, generate large reflectance anisotropy. The second source of difficulty is the regularization of the retrieval inverse problem, which frequently requires a priori knowledge on both the surface reflectance (e.g. spectral relationships, surface reflectance database) and the

aerosol optical properties (e.g. particle size distribution and refractive index) (Kaufman et al., 1997b ; Dubovik et al. 2011; Levy et al., 201; Hsu et al., 2019; Li et al., 2019).

Prior to the retrieval, several brightness and variability tests are generally applied to the TOA reflectances of selected spectral bands to screen out residual clouds, sediment contamination and non-optimal surface pixels. Then AOD is generally retrieved by minimizing the residuals between the TOA reflectances measured by the satellite for a given sun and satellite geometry and

the theoretical values which have been pre-computed from a radiative transfer model (RTM) for a set of candidate aerosol models and stored in a Look Up Table (LUT). Over ocean, the surface reflectance is generally computed from an ocean surface reflectance model which explicitly represents the contributions from the sun glint and the whitecap as a function of wind speed, and the reflection from within the water (Limbacher et al., 2014; Sayer et al., 2017; Garay et al., 2020), and AOD is retrieved independently. Over land, both surface reflectance and AOD can be simultaneously retrieved if the measurement information

content is high enough (Fougnie et al., 2020)

Retrievals are generally associated with i) a quality assessment (QA) flag which quantifies the overall confidence in the retrieval and is computed a posteriori from tests on the inputs and outputs of the retrieval algorithm  and ii) an error which can be a prognostic output from the optimization algorithm or a diagnostic computed a posteriori by evaluating the retrieval against ground observations (Sayer et al., 2020).




## A.2 Satellite retrieval algorithm

We provide below a summary of the retrieval algorithm of each product. The validation statements given for each product were taken from validation reports and relied on distinct methodology and different spatial and temporal sampling.

### A.2.1 MODIS Dark target (ocean and land)

AOD is retrieved over a 10 by 10 MODIS pixel retrieval box (~10 km at nadir) from the MODIS TOA reflectances which have been averaged over the retrieval box.

Over ocean, the surface reflectances in 6 spectral wavelengths (0.55, 0.65, 0.86, 1.24, 1.63, and 2.11 µm) are computed for various combinations of fine mode (selected from 4 models) and coarse mode (selected from 5 models) aerosols, which are characterized by a single mode log normal size distribution and a spherical shape. Since collection 6.0, the ocean surface
reflectance exploits a varying wind speed, taken from the NCEP forecast.

Over land, four aerosol models are prescribed as a function of location and season. This includes three fine-mode dominated models, which are characterized by a bi-log-normal size distribution, a spherical shape and distinct single scattering albedo, and a dust coarse-mode dominated model, which is bi-lognormal and non-spherical. The solution is a combination of the dust and one of the fine-mode models. Spectral relationships between the bands at 0.47, 0.65 and 2.11 µm, which are function of
NDVI and the scattering angle, are used to constraint the algorithm over vegetated areas. When not enough samples have been selected within the 10 by 10 retrieval box, an alternative retrieval is triggered using only the continental aerosol model characterized by a 3-mode log-normal size distribution and a spherical shape.

Each output is associated with i) a QA information derived from tests on the number of pixels selected within the 10 by 10 retrieval box and the degree of realism of the solution, and ii) a diagnostic error which has been computed as a function of
AERONET AOD (Table 1).

TERRA/MODIS DT was shown to be frequently the highest over open ocean conditions (Zhang et al., 2017, Sayer et al., 2018, de Leeuw 2018, Sogacheva et al., 2020). It has a positive offset at low AOD which scales with AOD and is mainly related to the calibration degradation of the TERRA/MODIS blue band (Levy et al., 2013; Sayer et al., 2018; Sogacheva et al., 2019). Validation results for collection 6.1 are reported at https://darktarget.gsfc.nasa.gov/validation/results. For AQUA/MODIS the
bias and the RMSE evaluated against AERONET are 0.023 and 0.096 over ocean and 0.013 and 0.1 over land. For TERRA/MODIS the bias and the RMSE are 0.039 and 0.099 over ocean and 0.029 and 0.106 over land. The percentage of samples within the expected error range are 83% and 76% over ocean and land, respectively, for AQUA and 77% and 73% over ocean and land, respectively, for TERRA.

### A.2.2 MODIS Deep blue (land)

The deep blue algorithm was first implemented in MODIS Collection 5 to fill in the dark target gaps over bright land surfaces (Hsu et al., 2013). Since collection 6.0, an enhanced DB algorithm, which includes updated cloud detection and modified



aerosol models, is applied to both bright and vegetated areas. Additional modifications were applied in collection 6.1, which includes updated radiometric calibration to L1b radiances, improved internal smoke detection, improved surface reflectance database over rugged and elevated terrain, updated parameters of the pixel level uncertainties (more details at

https://atmosphere-imager.gsfc.nasa.gov/sites/default/files/ModAtmo/modis_deep_blue_c61_changes2.pdf). Conversely to DT, DB applies the corrections for the radiometric degradation of the 0.412 and 0.470 μm blue bands of TERRA/MODIS. DB retrieval is first performed at 1km and then the 1km retrievals are averaged over a 10 by 10 MODIS pixel box. DB algorithm retrieves AOD and the fraction of two aerosol models from the radiances in the 0.412 μm and 0.47 μm spectral bands (Hsu et al., 2004). Distinct paths are used to estimate the surface reflectance depending on the surface type:

885        a.   Over vegetated surfaces, the surface reflectances in the blue (0.47 μm) and the red (0.65 μm) are estimated using spectral relationships between these bands and the SWIR (2.1μm). These relationships were derived from collocated MODIS observations with AERONET data which were stratified by geometry, land cover types (cropland and natural vegetation), season and vegetation amount quantified by a vegetation index (NDVI).

890        b.   Over bright surfaces (desert and mountains), a database of surface reflectance was derived from 7 years of MODIS data for each season and for different ranges of NDVI. The surface reflectance is parametrized as a function of the scattering angle to account for the non-Lambertian properties of the surface.

    c.   Over urban and cropland transitional regions, to account for the strong surface heterogeneity and anisotropy, the angular shapes of the surface BRDF were derived from collocated AERONET and MODIS

895            measurements, for distinct seasons and ranges of NDVI values (Hsu et al., 2013). The derived angular shapes are then combined with the surface reflectance values derived from the surface reflectance database at a scattering angle of 135°.

10 fine-mode and 5 coarse-mode aerosol models with spherical shape are employed in the retrieval. The size distributions and single scattering albedo of the fine-mode models are region-dependent. Since collection 6.0, they represent smoke and weakly-

absorbing aerosols to cover vegetated areas. Coarse-mode models employ the same phase function but have distinct single scattering albedo (Hsu et al., 2004). Since collection 6.0, MODIS infrared channels are used to identify extremely absorbing mineral dust prior to retrieval. An AOD is independently retrieved at each spectral band by selecting a single aerosol model. Then AOD at 0.55 μm is derived from the estimated AOD spectral dependence.

Each retrieval is associated with a QA based on residual cloud contamination, scene heterogeneity and number of retrieved

AOD pixels within each 10 by10 retrieval box (Sayer et al., 2013). A pixel level uncertainty, defined as one standard deviation Gaussian confidence interval, is computed from linear functions of MODIS AOD and solar and view geometry (Sayer et al., 2013). The parameters of the expected error for collection 6.1 can be found at https://atmosphere-imager.gsfc.nasa.gov/sites/default/files/ModAtmo/modis_deep_blue_c61_changes2.pdf





The evaluation of MODIS DB against AERONET measurements showed a bias less than 0.01, an RMSE of 0.012, 80% of retrievals within the algorithm expected error and 45% of retrievals within the GCOS uncertainty requirement (Sayer et al, 2018). The bias is generally small for background aerosol (AOD less than 0.2) and the negative bias increases from fine mode to dust. Regionally, performances are lower over biomass burning regions in South Africa, mixed polluted and dust sites in India, China, South East Asia, and desert sites where AOD is frequently underestimated (Tao et al., 2017, Sayer et al., 2018).

**A.2.3 VIIRS NOAA EPS**

The NOAA EPS NRT AOD product v2r1, derived from the Visible/Infrared Imager Radiometer Suite (VIIRS) on board the Suomi National Polar-orbiting Partnership satellite (SNPP) and the NOAA20 platform, is provided at the native pixel size of 0.750 km (Laszlo and Liu, 2020). Several internal tests are applied to the input TOA reflectances to filter out residual cloud-contaminated observations, sea ice, shallow water, glint and to identify heavy smoke and dust aerosols. The algorithm exploits
the 6S-V1.1 radiative transfer model to account for aerosol extinction, molecular scattering and gas absorption, and to couple the surface with the atmosphere. Final retrievals are categorized into four quality assurance levels based on internal tests and the retrieval residuals (Laszlo and Liu, 2020). Only best quality retrievals are selected for this work.

Over land, the TOA radiances at 0.412 μm (M1), 0.445 μm (M2), 0.488 μm (M3), 0.672 μm (M5) and 2.25 μm (M11) are used. The algorithm estimates the surface reflectances from the RED (M5) or the SWIR (M11) TOA reflectances because,
compared to the shorter wavelength bands, these bands have lower sensitivity to atmospheric scattering and higher sensitivity to surface reflectance. AOD is generally retrieved from the M3 blue band where the aerosol signal is strong, the surface is dark, and because this band is close to the nominal wavelength (0.55μm) where AOD is reported. The aerosol model corresponding to the retrieved AOD is selected using residuals, which are the departures of the TOA reflectances calculated at the rest of the spectral bands for a finite number of candidate aerosol models from the observed reflectances. For dark vegetated
surfaces, linear spectral relationships between M5 and M11, M3 and M5, M2 and M3, M1 and M3, and M11 and M5 were pre-computed for distinct land cover types as a function of NDVI$_{SWIR}$, the M5/M4 TOA reflectance ratio and the glint angle. For bright surfaces, the surface reflectance ratios with M5 are parameterized as a linear function of the scattering angle (using distinct parametrization for forward and backward geometries). They are derived at global scale from a static database at 0.1 ° spatial resolution which was computed using two years of VIIRS TOA reflectances over bright surfaces (Zhang et al, 2016).
Over North Africa and the Arabian Peninsula regions, a dust aerosol model is selected, the M3 spectral band is used to retrieve AOD and the residuals are calculated from M1 and M2. Over the rest of bright regions, AOD is retrieved from M1, which is better suited for AOD retrieval than M3 over bright surfaces, and the residuals are calculated using M2 and M3. The algorithm employs 4 aerosol models, namely generic, smoke, and urban fine mode-dominated models and a dust coarse mode, which are all characterized by a bi-modal lognormal aerosol size distribution, spherical shape for the fine mode models and a spheroid
shape for the dust model. These models are essentially based on the Collection 6 MODIS DT models. However, unlike the MODIS DT algorithm, which assigns the models to distinct geographical regions, the NOAA EPS algorithm dynamically selects the aerosol model based on the value of the residual.



Over ocean, the retrieval employs the 0.555 µm (M4), 0.672 µm (M5), 0.746 µm (M6), 0.865 µm (M7), 0.1240 µm (M8), 1.610 µm (M10) and 2.25 µm (M11) spectral bands. A typical model of ocean surface reflectance, which represents the

contributions from bi-directional sun-glint, Lambertian dark underwater and whitecap reflections is exploited. 5 coarse mode and 4 fine mode candidate aerosol models with spherical shapes (adopted from Remer et al., 2006) are used. Combination of the fine and coarse modes corresponding to varying fractions results in a large number of candidate aerosol models. AOD for each combination of fine and coarse mode is estimated using the M7 channel because of its low sensitivity to underwater reflectance and sufficient sensitivity to aerosols. The residuals at the rest of the spectral bands are used to select the best aerosol

model. The outputs are the fine and coarse mode aerosol models, the fine-mode fraction and the total AOD.

Parametric formulations of pixel-level uncertainty were derived from a posteriori evaluations against AERONET over land and ocean. Conversely to MODIS DB, no Gaussian assumption on the error distribution is applied and the expected error is estimated from the adjustment of the bias and the error variance as a function of the VIIRS AOD (Huang et al., (2016).

A first evaluation against AERONET for the period from October 2012 to March 2016 indicates bias and error standard

deviation of 0.01 and 0.1, respectively, over land and 0.03 and 0.05 over ocean (Laszlo et al., 2018). The ATBD (Table 2.1, Laszlo and Liu, 2020) provides accuracy and precision of AOD retrieval for three AERONET AOD ranges over land and two over ocean.

### A.2.4    Copernicus NRT SLSTR

The Optimized Simultaneous Surface-Aerosol Retrieval for Copernicus Sentinel-3 (OSSAR-CS3) is the reference EUMETSAT processor generating the NRT aerosol product, including AOD at 0.55 µm, derived from the radiances of the Sea and Land Surface Temperature Radiometer (SLSTR) dual-view instrument, on board Sentinel 3-A and B (Chimot et al., 2021). The collection 1.0 released in August 2020, which was available at the time of this work, is evaluated over ocean only. The following also includes a description of the land algorithm implemented in collection 2.0 (released in October 2021).

Prior to aerosol retrieval, absolute, inter-band and dual-view calibration corrections are applied to reduce the SLSTR radiance calibration uncertainties. The original L1B cloud mask is used over ocean while a specific cloud mask was designed to correct for under-screening deficiencies in the current L1B cloud mask over land. The aerosol product is provided at 9.5 km spatial resolution by aggregating a block of 19 by 19 native SLSTR pixels to reduce the impacts of surface heterogeneity, to mitigate co-registration errors across views and to decrease the retrieval computing time. The retrieval is triggered if only more than

50% of any of the 19 by 19 SLSTR pixels within each block are cloud and glint-free. A posteriori quality control tests, which include AOD spatial variability, residual of the spectral fit and reflectance brightness tests, are applied to flag AOD retrievals possibly contaminated by cloud and snow or ice residuals, sediments in coastal areas or impacted by other sources of uncertainties such as high ocean colour signal and bright surfaces in case of unfavourable geometries. A prognostic uncertainty (one standard deviation) is computed at the pixel level from the second derivative of the cost function at the optimal AOD.





OSSAR-CS3 employs the 6SV RTM to compute the surface reflectance for a set of pre-defined aerosol models which are a
linear combination of two coarse modes (sea salt, desert dust) and two fine modes (weakly and strongly absorbing). A spherical
particle shape is assumed except for dust which is modelled as a spheroid particle (Dubovik et al., 2006). Over ocean, the
retrieval relies on only the spectral information content of S2, S3, S5 and S6 spectral bands from all available views which are
used as independent spectral observations. The surface reflectance is pre-calculated using a ocean BRDF model, which
includes contributions from glint, white foam and ocean colour and uses the wind speed from the ECMWF forecast. Over land,
the S1, S2, S3, S5 and S6 spectral bands are exploited. Spectral weighs have been applied to the SLSTR radiances to limit the
impacts of their uncertainties on the retrieval for given surface types and geometry configurations. The retrieval algorithm is
a combination of the North et al., 1999 dual-angular model, used in a joint aerosol-surface reflectance fit, and a spectral first
guess for the RED surface reflectance derived from the NIR or the SWIR radiances (Chimot et al., 2021). The weights between
both approaches are a function of land surface type and dual-view geometry configuration. The dual-angular model is favoured
for low scattering angles and over bare soils while the spectral constraint is required to compensate for the uncertainties of the
dual-angular model at large scattering angles (>110°) and over developed-vegetation.

Level-2 AOD evaluation results (Chimot et al., 2021) against 1.5 year of AERONET observations showed good performances
of the collection 1 over ocean with a correlation of 0.9, a bias between -0.01 and -0.03 for AOD <0.1, an RMSE of 0.06 and a
compliance with GCOS uncertainty between 66% and 72%. Land collection 2 retrieval showed lower performances compared
to ocean with correlation, bias, RMSE and a GCOS fraction of 0.77, 0.061, 0.169 and 29%. Correlation with AERONET is
lower at low AOD in case of unfavourable geometry for which the discrimination between the low aerosol signal and the high
surface reflectance is largely uncertain.


### A.2.5 PMAp

The Polar Multi-Sensor Aerosol Product (PMAp) is derived from observations from the combined use of GOME-2 UV-VIS
spectrometer, the IASI Fourier transform infrared sounding interferometer and the AVHRR radiometer on board Metop-A,
Metop-B and Metop-C (Grzegorski et al., 2022). The PMAp v2.1 data set from METOP-A and METOP-B, which was available
at the time of this work, is used in this work. A new version of PMAp (2.2.4) has been recently released and has been used in
the operational CAMS system since July 2021.

PMAp is produced at the spatial resolution of GOME-2 (5 x 40 km² for Metop-A and 10 x 40 km² for Metop-B). The linearly
polarized radiances measured by the GOME-2 Polarisation Measurement Devices (PMDs) are used to derive unpolarized and
polarized TOA reflectances which both are the inputs of the AOD retrieval algorithm. A radiometric correction scheme was
implemented to account for the spectral degradation of GOME-2 reflectances due to the ageing of the instrument.

AVHRR observations are used for cloud detection, cloud fraction estimation and cloud correction calculation. A preliminary
aerosol optical properties classification is performed prior to retrieval. This includes i) computation of a dust index exploiting
the IASI infrared thermal spectra, ii) identification of volcanic ash exploiting both AVHRR and IASI observations and GOME-





2 UV index and iii) a fine/coarse mode discrimination derived from AVHRR spectral ratios. The retrieval relies on separate
LUTs for ocean and land, which contain the reflectances and stokes fractions for ten PMD bands and various aerosol models
(up to 29 aerosol models but in the current version 9 models are used for ocean and 5 for land). Over ocean, the surface
reflectance is pre-computed in cloud-free conditions using ECMWF wind speed forecast and estimated chlorophyll
concentration. A default chlorophyll concentration value is used for partially cloudy pixels. Over land, the algorithm exploits
a priori information on surface reflectance derived from the GOME-2 Lambertian-Equivalent reflectance monthly climatology
(Tilstra et al., 2017). The angular dependency of surface reflectance is accounted for using GOME-2 viewing angle information
content (Tilstra et al., 2021). The candidate aerosol models identified in the pre-processing step are used to derive a series of
AOD estimates using unpolarised GOME-2 reflectance at specified channels for ocean (0.650 μm) and land (0.410 μm and
0.470 μm). Both un-polarized and polarized reflectances are used in the optimization process to retrieve AOD.

QA information is computed depending on wind speed, geometry configuration, aerosol type pre-classification, cloud and
thick aerosol screening, quality of fit and range of AOD values. A prognostic error is computed as one standard deviation of a
set of a minimum 30 AOD estimates obtained using perturbations of selected input parameters of the retrieval algorithm.
Additional information on PMAp retrieval can be found at https://www.eumetsat.int/media/39243.

The validation of PMAp v2.1 against AERONET within the June-Sept 2013 and Feb-May 2015 reference periods indicated a
better correlation over ocean (~0.8) than over land (~0.6). Over ocean, the range of slope and offset of best fit line are
0.5-0.8 and 0.04-0.1, respectively. Over land, the range of slope and offset of best fit line are 0.5-0.7 and 0.1-0.2,
respectively (Table 3 and 4 on page 19 of PMAp validation report available
at https://www.eumetsat.int/media/40632)






## Appendix B: Product scatterplots


**Figure B.1: AOD satellite scatterplots of temporal mean AOD over ocean, for the DJF period**






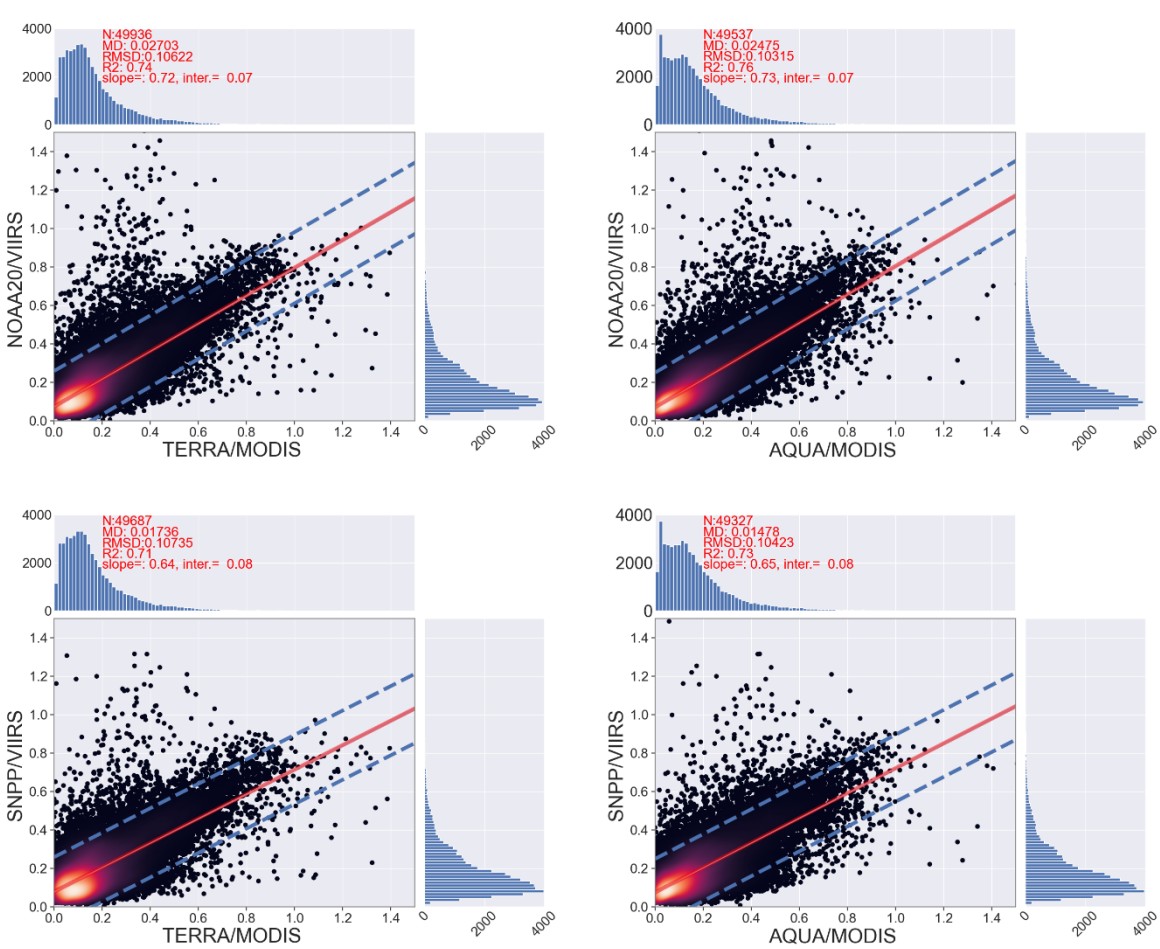


**Figure B.2: AOD satellite scatterplots of temporal mean AOD over land, for the DJF period**





**Appendix C: Regional histograms**

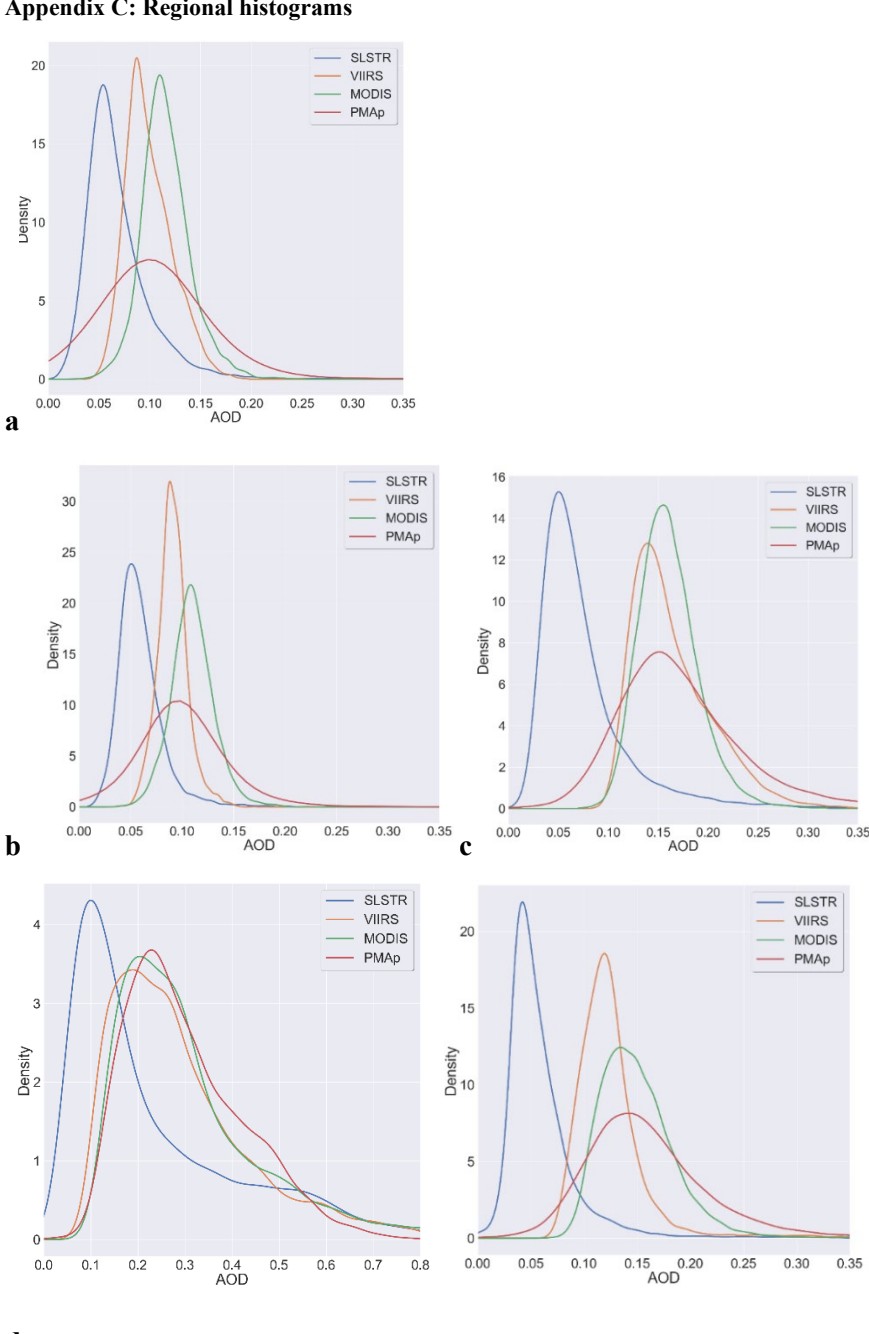

d e


**Figure C.1: AOD distributions over distinct ocean regional domains for the DJF period. a: North Atlantic, b: North Pacific, c: South Pacific, d: Mid-Atlantic, e: South Indian ocean**






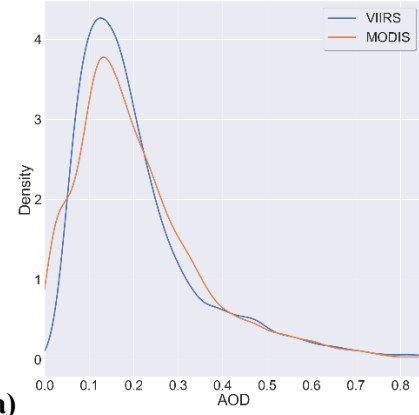

a)

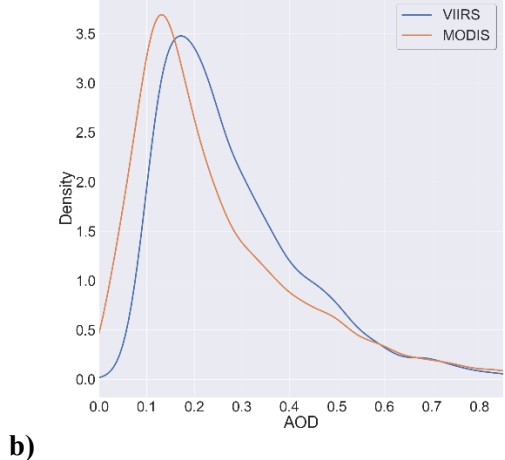

b)

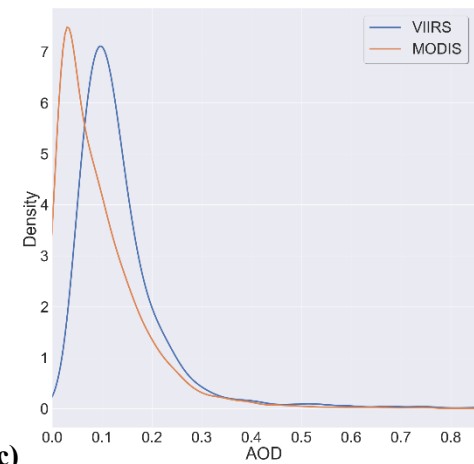

c)





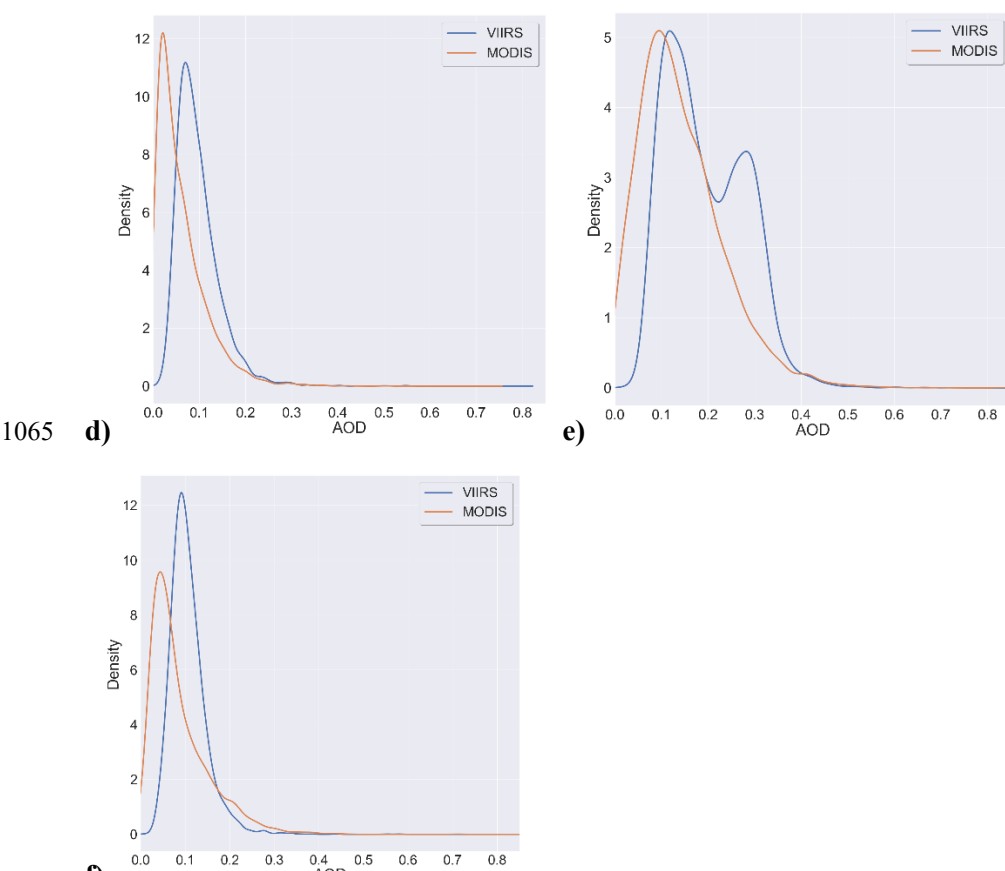

**d)**

**e)**

**f)**

**Figure C.2: AOD distributions over distinct land regional domains for the DJF period. a: Africa, b: Asia, c: Australia, d: North America, e: South America, f: Europe**





**Appendix D: Additional results for the MAM period**

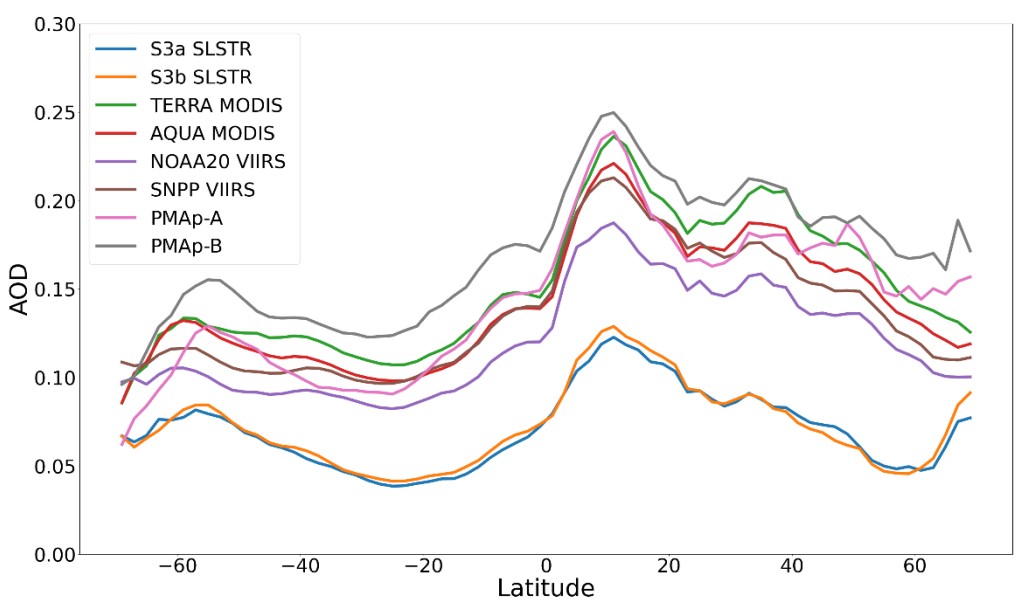

**Figure D.1 Latitude cross-section of temporal mean satellite AOD, for the MAM period over ocean.**






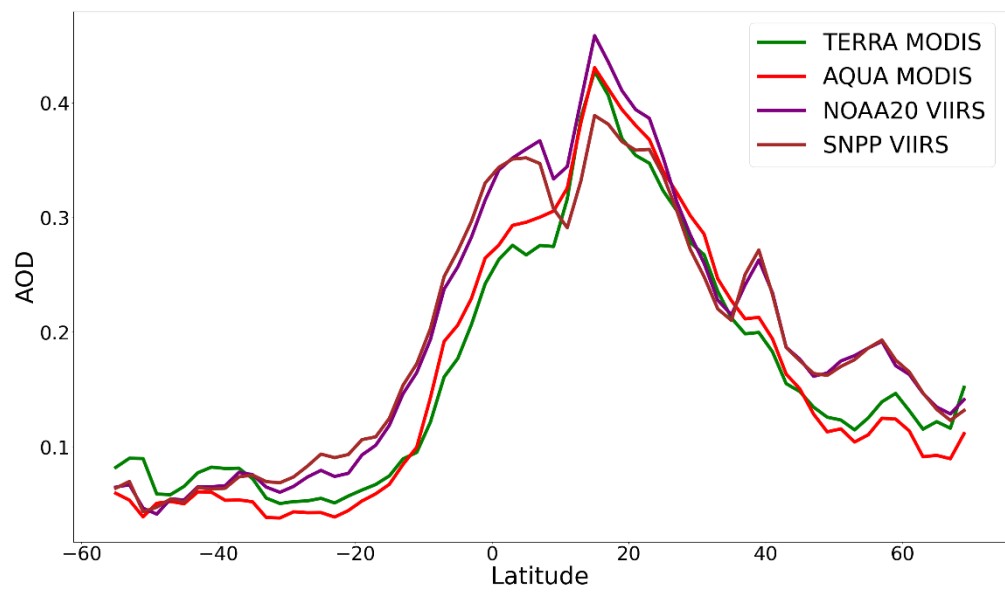

**Figure D.2** Latitude cross-section of temporal mean satellite AOD, for the MAM period over land.





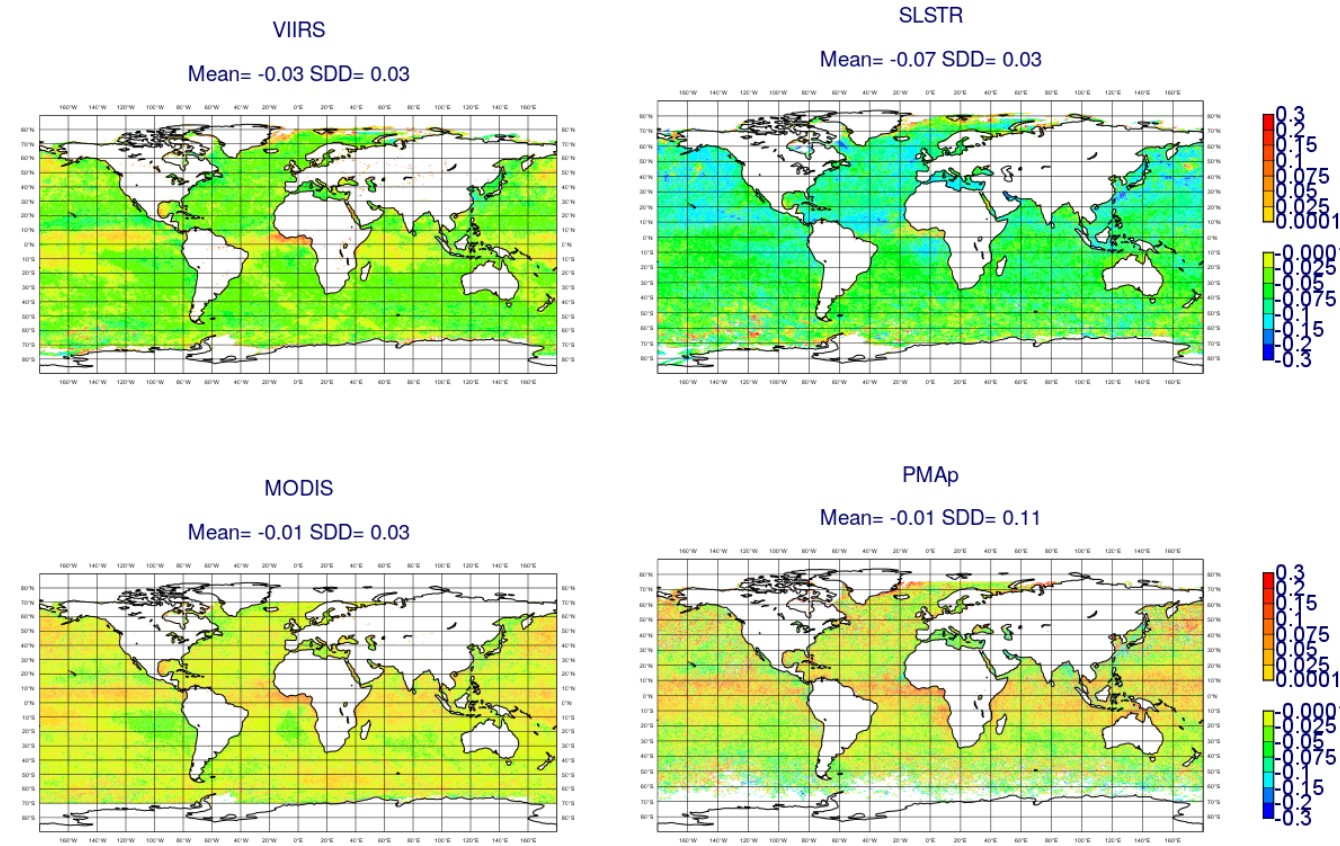


**Figure D.3 Global maps of mean of first guess departure from TERRA&AQUA/MODIS, NOAA20&SNPP/VIIRS, S3A&S3B/SLSTR, METOP-A&B/PMAp for the MAM period over ocean**





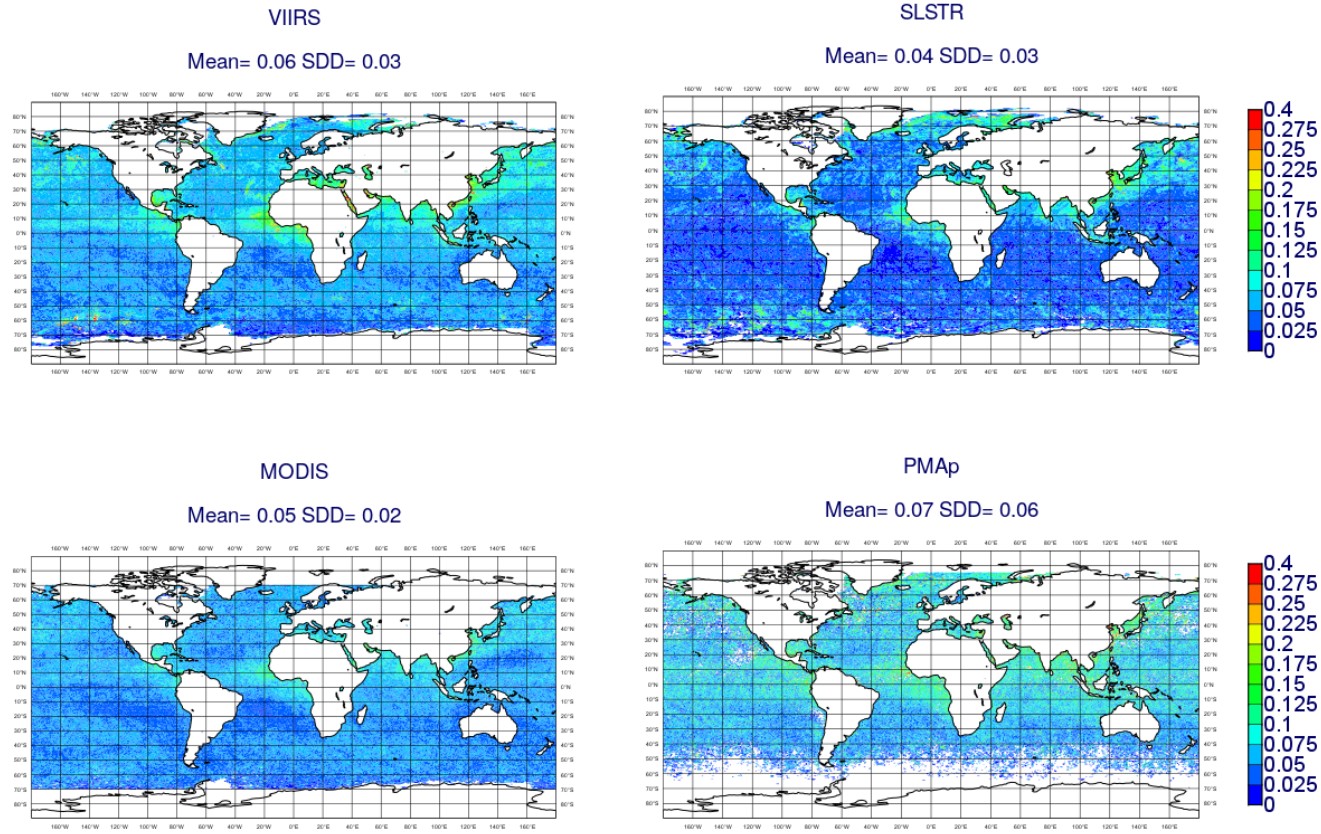

**Figure D.4|: Global maps of the standard deviation of first guess departure from TERRA&AQUA/MODIS, NOAA20&SNPP/VIIRS, S3A&S3B/SLSTR, METOP-A&B/PMAp, for the MAM (2019-2020) period, over ocean.**






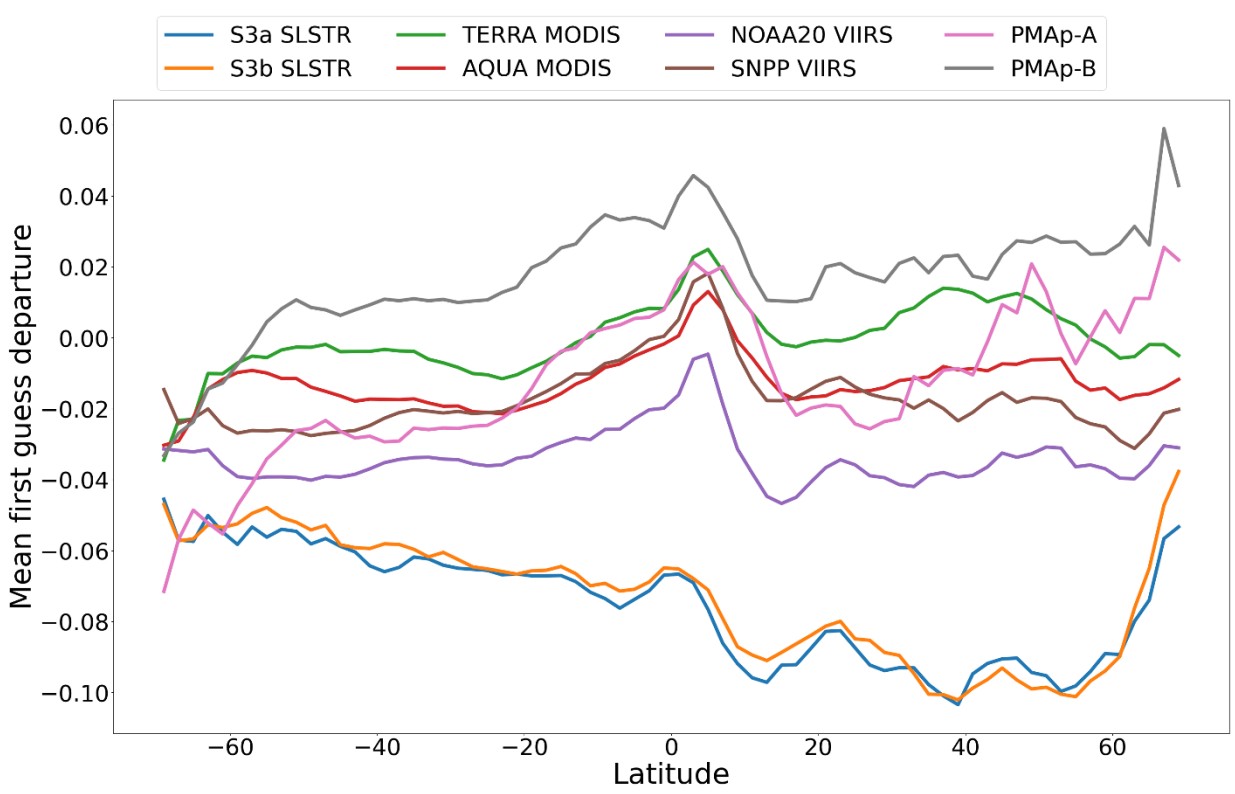

**Figure D.5: Latitude cross-section of first guess departure mean, for the MAM (2019-2020) period, over ocean.**






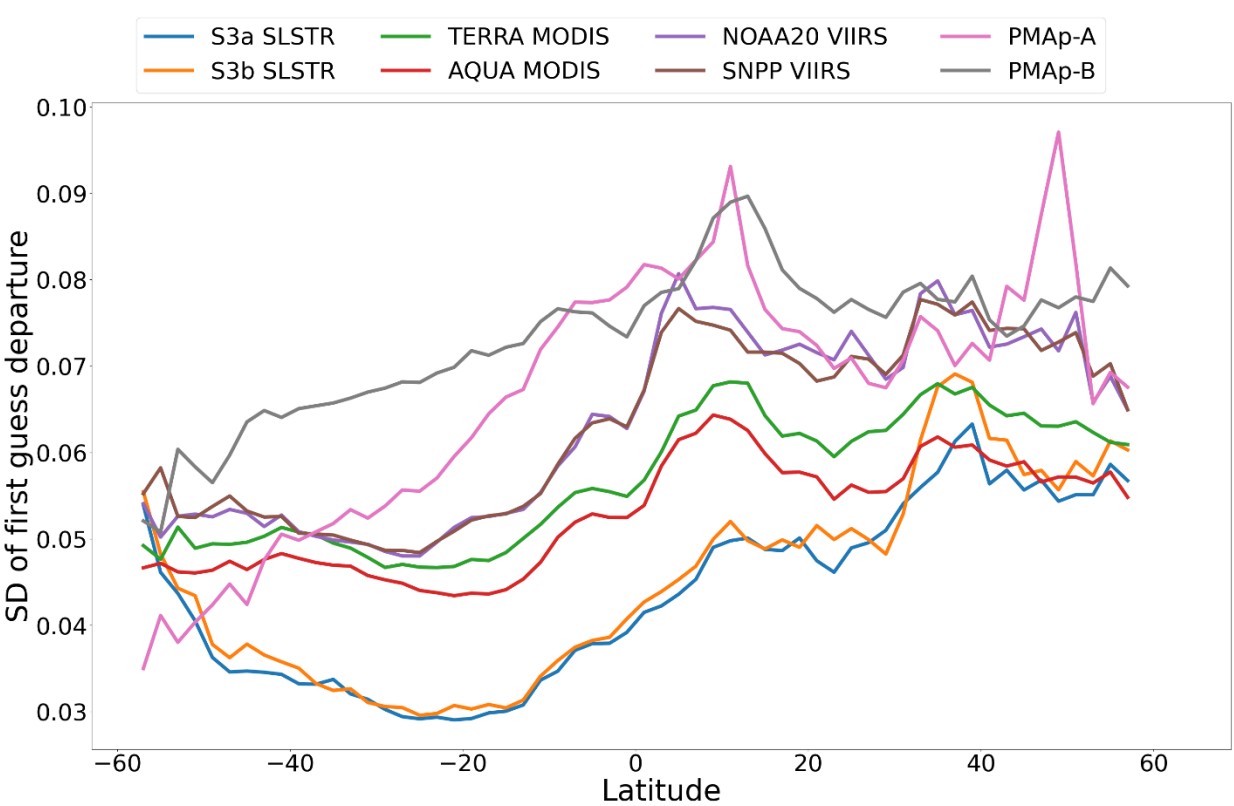


**Figure D.6: Latitude cross-section of first guess departure standard deviation (SD), for the MAM (2019-2020) period, over ocean.**







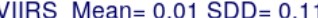

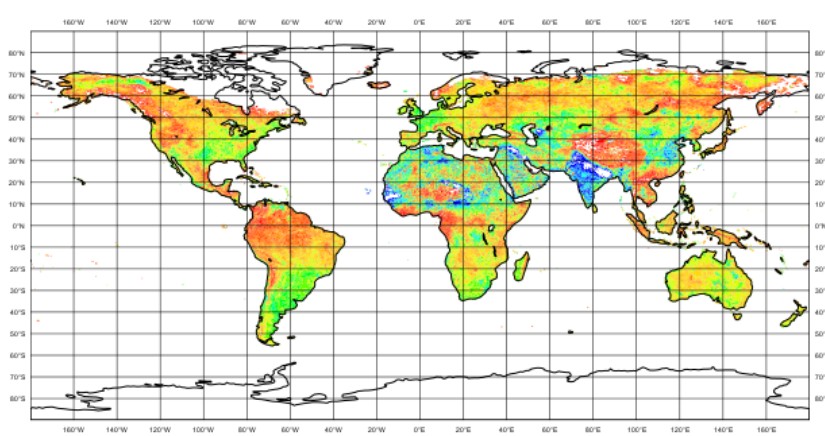

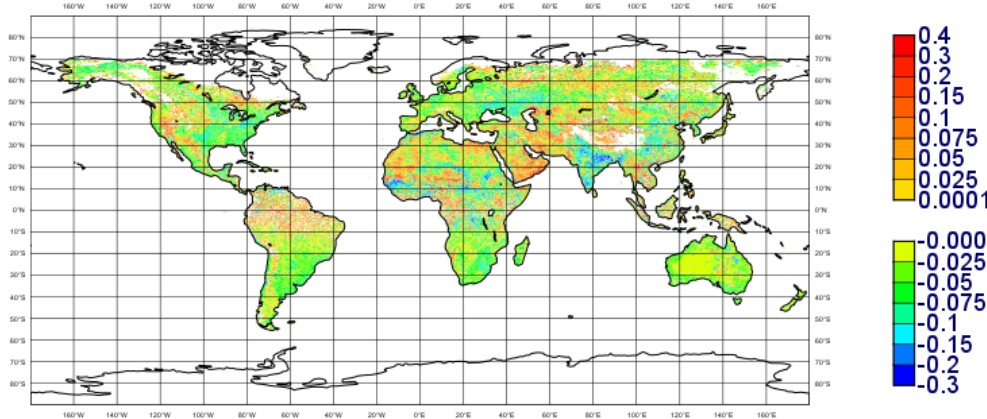


**Figure D.7: Global maps of mean first guess departure from TERRA&AQUA/MODIS, NOAA20&SNPP/VIIRS for the MAM period over land**




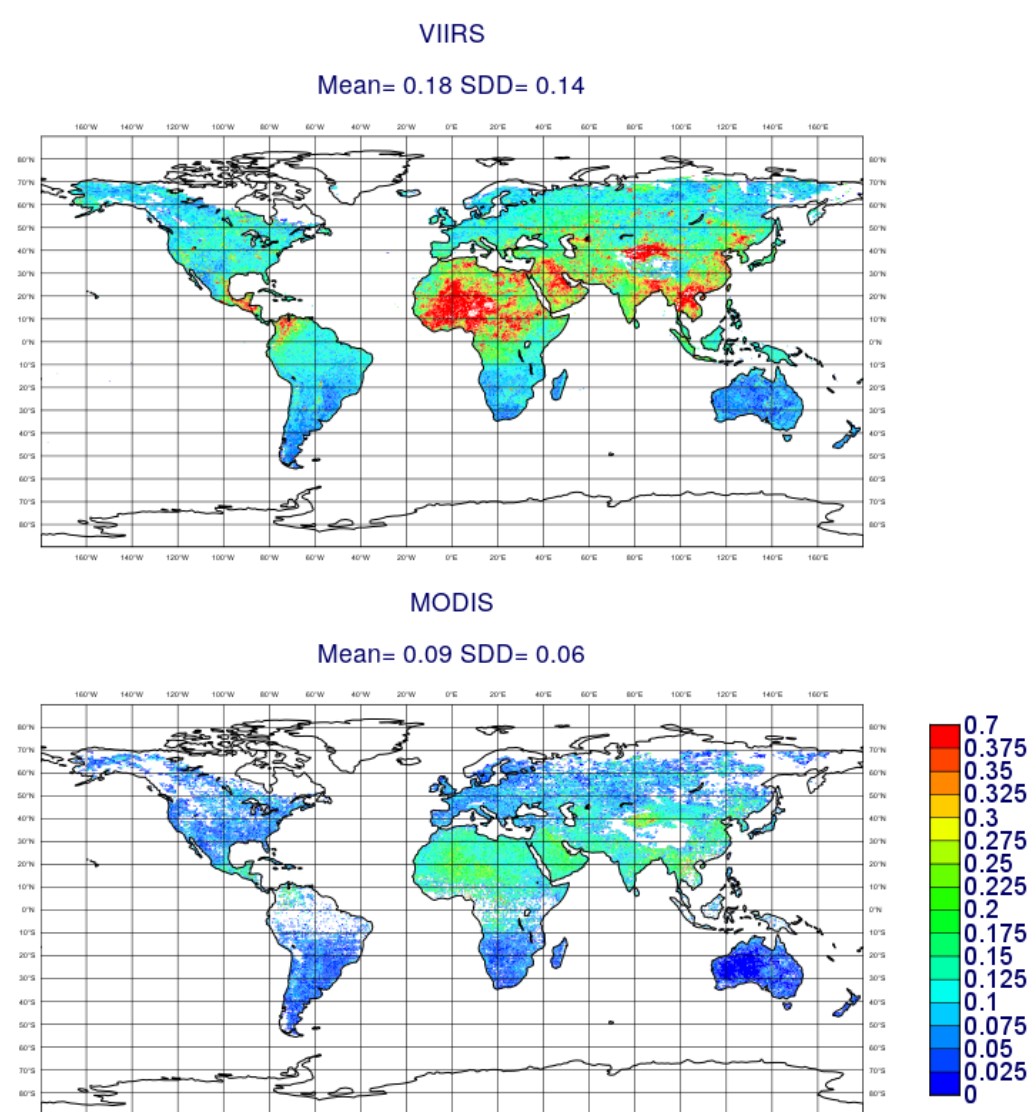

**Figure D.8: Global maps of standard deviation of first guess departure from TERRA&AQUA/MODIS, NOAA20&SNPP/VIIRS for the MAM period over land**







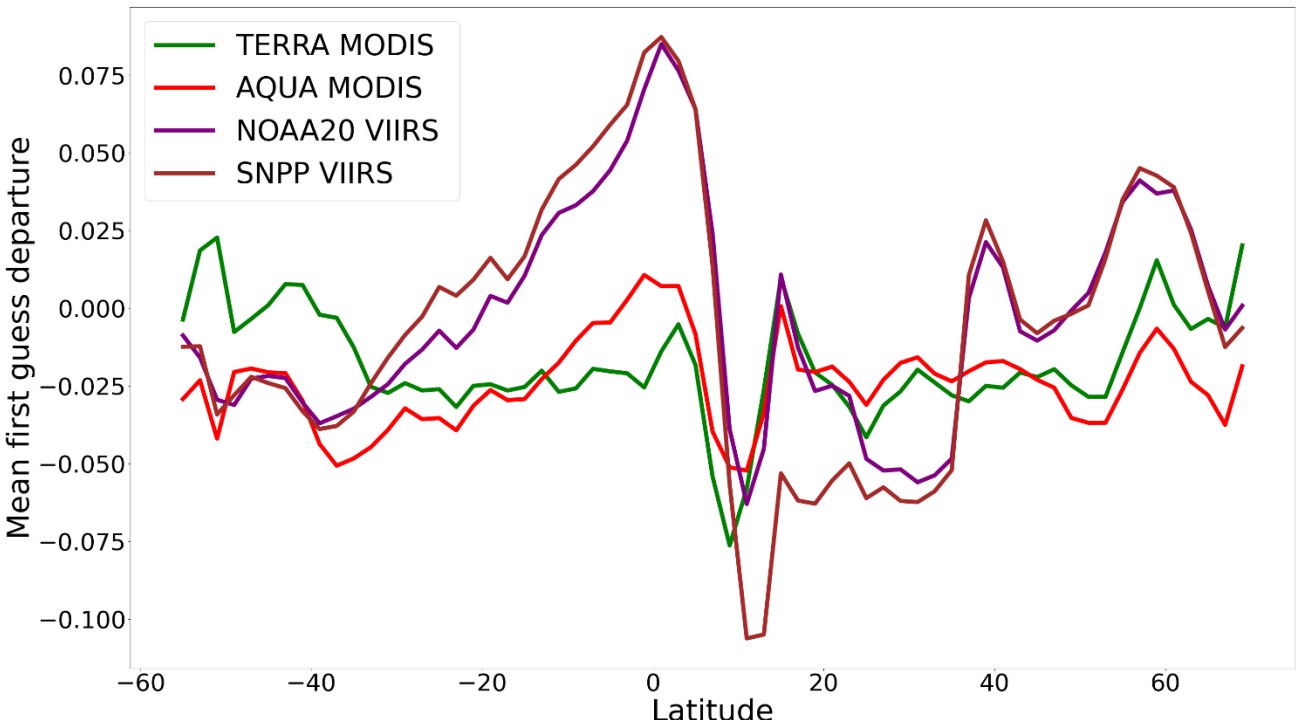

**Figure D.9: Latitude cross-section of mean of first guess departure, for the MAM (2019-2020) period, over land.**







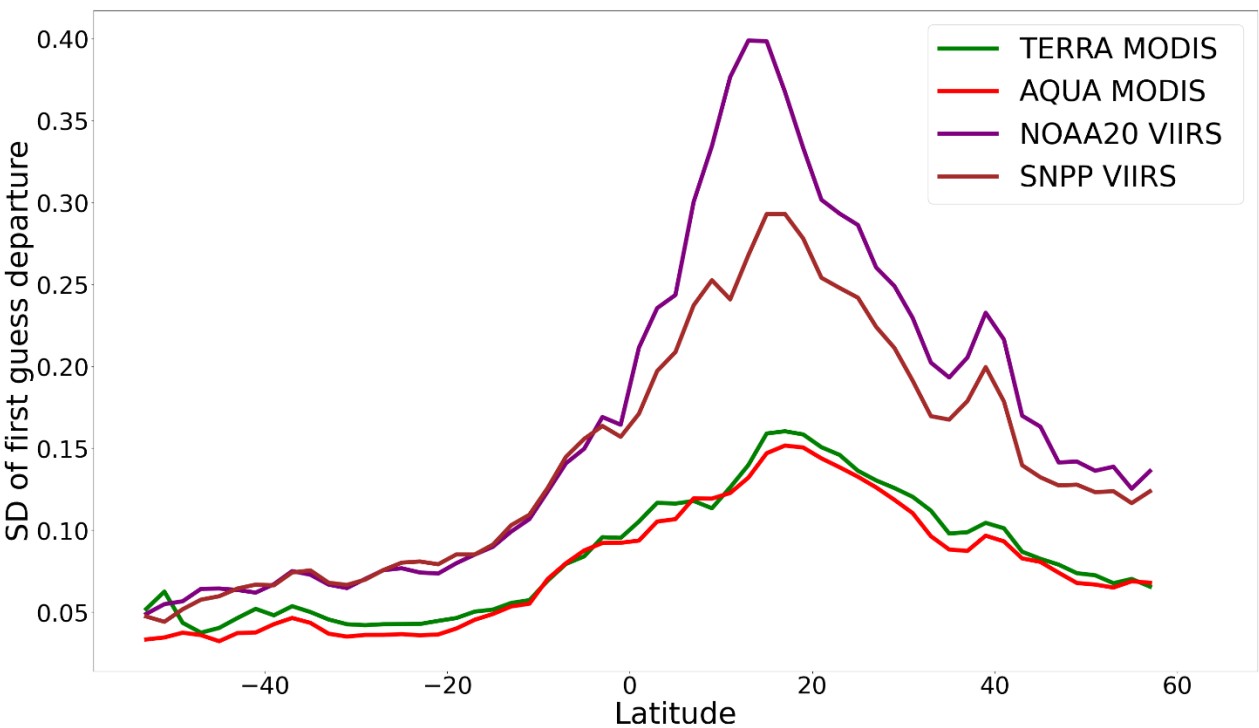


**Figure D.10: Latitude cross-section of SD of first guess departure, for the MAM (2019-2020) period, over land.**






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
