# Peer review of "Monitoring multiple satellite Aerosol Optical Depth (AOD) products within the Copernicus Atmosphere Monitoring Service (CAMS) data assimilation system"

_Atmospheric Chemistry and Physics, 2022_

## Author Response (AR1)

Dr Sebastien Garrigues
Copernicus Atmosphere Monitoring Service (CAMS)
European Centre for Medium-Range Weather Forecasts (ECMWF)
Shinfield Park, Reading, RG2 9AX, United Kingdom
E-mail: sebastien.garrigues@ecmwf.int
Tel: +44 (0) 118 949 9264

Reading, 08/07/2022,

Answer to final review of "Monitoring multiple satellite Aerosol Optical Depth (AOD) products within the Copernicus Atmosphere Monitoring Service (CAMS) data assimilation system" as a research paper for ACP.

Dear Editor,

Please find the revised version of the manuscript entitled "Monitoring multiple satellite Aerosol Optical Depth (AOD) products within the Copernicus Atmosphere Monitoring Service (CAMS) data assimilation system" along with the response to the reviewer's comments posted in ACPD.

Section A provides the responses to the first quick review (reviewer 1 and 2) which was made prior to publication in ACPD.

Section B presents my answers to reviewer 1 and 3 comments posted in the discussion forum. Changes in the revised manuscript are highlighted in yellow.

We would like to thank all reviewers for their valuable feedbacks on the paper that helped to improve the quality of the paper.

**A/ Response to reviewer 1 and reviewer 2 quick review prior to discussion in ACPD**

**A.1 Response to reviewer 1**
1. **Paper edits**
   - The abstract was substantially reduced to focus only on the main outcomes of the work. It is reduced to 2 paragraphs with a total of 350 words (compared to 505 words in the submitted manuscript)
   - Introduction: we removed the generalities about importance of aerosols for climate and air quality. We agree that this is well known by the audience. We condensed the paragraph on aerosol modelling and aerosol retrieval. We added few statements on AOD data assimilation. The introduction was reduced by 700 words.
   - Section 2:
     - We substantially reduced this section (~700 words compared to ~3000 words in the previous version). We kept the main characteristics of the AOD products which are summarized in Table 1. We moved the detailed description of each retrieval algorithm in Appendix A. We think that it is valuable for the readers to have such a detailed description of NRT AOD retrieval algorithms grouped in a single paper. Besides, for VIIRS and SLSTR, no peer-review paper describes the last version of the retrieval

algorithm.
- Appendix A.1: we removed it. We kept some general statements to introduce the new appendix A on satellite retrieval description.
- Appendix A.2: We used some statements from section A.2 on AOD uncertainties to consolidate the introduction and the discussion.
- We reduced the Conclusion.
- We also conducted a thoroughly reading of the paper to reduce the text and improve it.

2. **Page 12, line 285**: No reference is available to support this statement. We thus decided to remove it.

3. **Page 16, line 397-399:** These lines mean a two-step "Level 2 products to instantaneous average on model 40 km grid to 3-month model 40 km average." So the analysis is relevant. We agree that the text was ambiguous, and we replaced it by the following:

> "The intercomparsison was carried out at the IFS model spatial resolution (~40 km) and at a 3-month temporal resolution. This was done in two steps: (1) instantaneous regridding of level-2 retrieval product at the (level-3) IFS model spatial resolution and (2) 3-month average of the instantaneous level-3 AOD retrieval."

4. inappropriate SDD acronym : SDD was replaced by SD as suggested

**A.2 Response to reviewer 2**

1. **Bias in CAMS AOD forecast and analysis:** We agree that the AOD analysis and forecast from CAMS are associated with various sources of uncertainties. In the paper, we do not aim to use the CAMS AOD analysis as an independent reference to evaluate the AOD retrieval from satellites. The satellite retrievals are compared to the model through the first guess departure which drives the assimilation. The first guess departure is used in this work to assess the inconsistencies between AOD satellite products with respect to their departure with the model. As suggested by reviewer 2, we acknowledge the sources of biases in the CAMS AOD predictions based on the CAMS validation report available for the 2019-2020 period (doi:10.24380/322n-jn39). We created a new subsection 3.2 on this aspect. In introduction, we also acknowledge the biases frequently associated with global aerosol models as highlighted in the Sessions et al., 2015 paper. We used the three references provided by reviewer 2.

2. **"Regarding the conclusion that VIIRS and MODIS show the best agreement between themselves and CAMS. keep in mind that it is not only algorithm, differences, but I expect MODIS drive the bulk of the DA system-so it should compare better to CAMS."** We agree that the assimilation of MODIS in CAMS reduces the departure between MODIS AOD and the model. This is shown by the smaller standard deviation of MODIS first guess departure (Fig 8 and Fig 15). This is clearly stated in the text: lines 385 (Section 4.1.2) for ocean and lines 545 (section 4.2.2) for land.

3. **"CAMS hysterograms and zonal plots along with the satellites."** The comparison with the CAMS model is achieved through the first guess departure which represents the differences between the level-2 retrieval at its native spatial resolution and the model-simulated equivalent observation with proper collocation in time and space. The mean and standard deviation of first guess departure presented in Fig. 5 for ocean and in Fig. 12 for land are meaningful estimates of the systematic and random, respectively, differences between the observation and the model that can be used to identify possible spatial inconsistencies between AOD satellite products. We think that the first guess departure plots convey enough information to understand how the observations compare to the CAMS first guess. As suggested by the reviewer, we strengthened the comparison of the satellite retrieval with the CAMS model by adding the latitude cross sections plots of the mean and the standard

deviation of first guess departure (new Fig. 7 and 8 for ocean and Fig. 16 and 17 for land). For clarity, we prefer to not include the model first guess in the zonal plots shown in Fig.4 and 11 which are dedicated to retrieval comparison.

4. **"Are you weighting your area averages by latitude/longitude or equal area? We get lower mean MODIS AODs than you for the global average over long timer periods."** The gridding used to produce the observation maps was achieved using equal latitude-longitude *square grid*. This leads to larger area toward the equator compared to higher latitudes. But this should not affect the visual comparison of the AOD maps between products. Besides, this source of representativity difference is likely negligible compared to other sources (cloud contamination, product native spatial resolution …). Regarding the differences of global average of MODIS AOD: we provide separate values for ocean and land and for the DJF and MAM periods. Global values are lower over ocean than land. We also obtain different values for the DJF and MAM periods. Global values can be different over longer period of time and also vary with MODIS data set collection.

5. **"I think doing two 3-month segments is fine for demonstrating the method, but if the job is going to be done right, they really need years of data. They may want to comment on that. "** The two contrasted 3-month periods (winter and spring) used in this work encompass different aerosol events (e.g. Australian fires for DJF, Sahara dust outbreak, …) that are representative of the global variability of aerosols. We agree that longer time periods would allow to better resolve the seasonal and interannual variability of aerosols and the associated meteorological conditions (e.g. transport). Further work should include the Northern hemisphere summer to better assess the products over the frequent North America and Siberia fires. We added a comment on this in Discussion (line 697).

**B/ Response to reviewer 1 and reviewer 3 posted in ACPD**

**B.1/ Response to reviewer 1**

1. *"Line 59: think this should be 2000, not 2001?":*

   We agree the starting year for MODIS data is 2000

2. *"Line 65: a citation for the GCOS requirements should be added. I think the numbers for some geophysical quantities change periodically":*

   This was taken from the GCOS implementation needs GCOS-200, 2016 report. We added the reference.

3. *"Table 1: the MODIS DT ocean uncertainty is missing from the table. I believe it is 0.04+10%*AERONET in Collection 6 onwards (based on Levy et al 2013 paper cited in the manuscript). " :*

   The MODIS DT uncertainty for C6.1 is included in Table 1. It was taken from the updated information given in the official DT website (https://darktarget.gsfc.nasa.gov/validation/results)

4. *"Section 2.1.1: might be good to specify here again that this is the NOAA VIIRS product. There are NASA DB and DT VIIRS products as well. I see people cite the NASA papers using the NOAA products sometimes, and vice versa, so doesn't hurt to add a sentence saying directly that NASA products exist but are not used here. I don't know the NRT status of the NASA VIIRS ones at present, and see it makes sense to use the NOAA ones given they are NRT (and the resolution is nice too)."*

As suggested by reviewer 1, we added a sentence in paragraph 2.1.1 to clearly state that the VIIRS product used in this work is the EPS dataset produced by NOAA in NRT. We acknowledge that VIIRS AOD datasets are also produced in NRT by NASA (Sayer et al., 2017; Hsu et al., 2019; Sawyer et al., 2020). However, the NASA dataset only include retrievals from S-NPP while the NOAA EPS product includes both S-NPP and NOAA20 retrievals. For this reason, it was decided to implement and test the NOAA product in CAMS. We may envisage to test the NASA products in the future (the next processing should include NOAA20).

5. *"Line 162: I think this is the first time the acronym TOA is used, I know what it means but it should be defined":*

   We defined TOA here.

6. *"Line 169: is there a paper or tech report citation about the issues with PMAp and SLSTR over land?":*

   We used the validation reports of PMAp (EUMETSAT, 2021a) and SLSTR (EUMETSAT, 2021b) to justify their lack of accuracy over lands.

7. *"Line 170: is this really how DB and DT are combined here? If so, why not just use the merged product provided within the files? It is not the same as gap filling DB with DT, there is some averaging and QA comparison done too. See the Levy (2013) paper mentioned earlier".*

   The combined DT-DB product has been available from NASA since collection 6.0. It was not available at the time of the operational implementation of MODIS DB in CAMS. It was thus decided to use best quality DT retrieval and to gap-fill it with best quality DB retrieval. We agree that this would be slightly different than using the combined product produced by NASA which consists in selecting the DB or DT retrieval which has the best QA value or averaging both if their QA are equal. We added a sentence to justify our choice.

8. *"Line 222: for non-modeler readers, it would be useful to state what the TL511 model resolution corresponds to in km or degrees. Is this the same as the 40 km resolution mentioned on line 230 or something different?"*

   TL511 is equivalent to a grid size of about 40 km. We added it at line 222.

9. *"Paragraph beginning line 248: if I understand correctly, the first guess departure will be useful for the absolute evaluation of the satellite products if the (un-assimilated) model itself is somewhat skillful. If the model is not good at a certain place/time then you wouldn't necessarily know whether the difference is due to model or observation errors, and conversely if the model were perfect you could use it perfectly to diagnose observation errors (but then assimilation itself wouldn't bring a benefit). Is that right? I suggest adding a sentence or two here for non-modeler readers to explain more why this is a useful metric and what the caveats/assumptions are. Presumably the fact that first guess departure is based on the model field including MODIS/PMAp assimilation from previous time steps, makes up for some potential errors in the model (assuming in that case that MODIS/PMAp in the previous time step were good)."*

   The first guess departure represents the differences between the observation and the model equivalent of the observed variable at the time and location of the observation (observation space). It is the result of both the observation and the model errors (bias and random error). The mean and the standard deviation of the first guess departure represent the systematic and

random, respectively, components of the difference between the observation and the model. Model errors mainly arise from uncertainties in process representation, parameters and forcing variables. Observation errors include retrieval error (e.g. errors in radiance measurement and retrieval algorithm) and representation errors (e.g. observation operator used to convert the model variable into its observation equivalent, spatial and temporal interpolation to map the model variable in the observation space). Data assimilation systems are designed to correct small changes and random errors. The first guess departure needs to keep reasonable small values to mitigate the impact of non-linearities. Any bias in the observation may lead to larger errors in the analysis and can result in inconsistencies between distinct satellite observations which may fight against each other when they are assimilated. The use of the first guess departure is twofold: i) check that the mean departure between each type of observation and the model is reasonably small and not impacted by any biases in the observation and ii) evaluate the retrievals relatively to the model to identify possible spatial and temporal inconsistencies between AOD satellite products that would impact the assimilation of multi-satellite AODs. This requires that the model is skilful to some extent and lowly biased compared to the observation which a reasonable assumption given that the short-range forecast used to compute the first guess departure is simulated from optimised initial conditions produced by the data assimilation system. We agree with the reviewer's remark that this is not a pure model-observation comparison since the simulated values are influenced by the previous assimilation cycles. Besides, lower mean and SD of first guess departure are expected for MODIS which is assimilated and influence the last analysis cycle. The characteristics and the role of the first guess departure and the lower first guess departure expected for assimilated observations are provided in the last paragraph of Section 3.3 of the submitted version. As suggested by reviewer-1, we added two sentences on the assumptions associated with the use of the first guess departure for the relative evaluation of multi-satellite AOD retrievals.

10. *"Figure 1, 2, 5, 6: the paper says the analysis is only for date with latitudes smaller than 70 degrees. The maps include data above 70 degrees (except for MODIS which seems to have a cutoff). So it's not clear if that data is used in the discussion of this figure or not. If data above 70 degrees are not used, I would suggest not plotting it in the maps."*

We agree and we modified the plot to display data within 70S-70N

11. *"Mapped figures: these still say SDD for standard deviation, not SD like in the rest of the paper. I also wonder if it's possible to put the mean and SD on the same line as the sensor name, with the two-line plot titles there's a lot of space between panels which makes it a bit harder to visually compare than if they were closer together."*

We replaced SDD by SD and we adjusted the title as suggested.

12. *"Line 374: I wonder if here (or earlier) you could introduce an acronym FGD or a symbol for "first guess departure". The phrase appears a lot in this section, it would be easier if there were some shorthand for this term. I counted 22 uses on page 22 alone, and about a dozen on the corresponding land section (page 24).*

"We use FGD as acronym for first guess departure.

13. *"Line 376: if I understand correctly, negative first guess departure means the satellite is lower than the model field. Is that right, or do I have it backwards? For a non-modeler reader, as this is the first example given in the paper, it would be good to state this clearly to make sure people don't get the conclusions backwards. If that is correct then would it imply the model AOD is higher over ocean (since we know most satellites are also too high) –*

*possibly because of the assimilation of biased MODIS and PMAp observations in the previous time step?"*

The first guess departure is the difference between the observation and the model. Negative first guess departure values indicate that the satellite AOD is lower than the value of the model. We explicitly state this in the text to avoid any misinterpretation (line 391 of the new MS). The first guess departure of VIIRS and SLSTR are frequently negative for the oceanic background aerosol. The larger value of the model can be related to i) positive bias in the model for sea salt and ii) the assimilation of TERRA/MODIS which is known to be positively biased ocean. This is consistent with the lower value of VIIRS AOD than MODIS AOD over ocean.

14. *"Figures 7, 16, D5, D9: it would be useful to add the horizontal line y=0 here, as a reference for zero mean departure."*

We added horizontal lines on Figures 7, 16, D5, D9.

15. *"Section 4.2: how are the unphysical negative AOD retrievals in the Dark Target land product handled here? Are they set to 0, set to invalid data, or something else? From the Levy (2013) paper again, it happens about 20% of the time over land (see e.g. Figure 10 of that paper). As a non-assimilation reader it would also be useful to know how these are handled in the assimilation process because of course a negative aerosol mass would not make sense. This should be explained as it's an important long-term issue with that data product which is relevant for assimilation."*

MODIS DT retrieval allows negative AOD retrieval to avoid artificial bias in long time series. To avoid unphysical AOD values, it was decided to set the negative AOD values from the DT retrieval algorithm to zero in the pre-processing of the MODIS observation (Benedetti et al., 2009)

16. *"Line 618: the Schugens reference should be Schutgens. "*

We corrected the reference

17. *"Line 620: Sea Surface Temperature does not need to be capitalized."*

We corrected sea surface temperature

18. *"Lines 644-646: This comment about resolution seems speculative and unsupported to me. At those transport distances, I don't see why MODIS would not see the transported smoke at 10 km retrievals but VIIRS would at 750 m. From looking at imagery of the event, the source plumes fairly quickly spread out to more diffuse areas tens to hundreds of km wide. I think it is more likely that the differences in the model-scale aggregate are influenced by different populations of high vs. low AOD retrievals (real or artefact) in these two products, i.e. a pixel screening issue, not a retrieval resolution issue. If the authors want to make a claim here, evidence should be shown to back it up. For example showing examples of the source L2 data and of the L3 data for such a transported smoke case would quickly show what pixels from each sensor are available and what the retrievals look like."*

The differences between MODIS and VIIRS AOD at the model spatial resolution with respect to the detection of the transported Australian smoke plume in the South Pacific are likely related to differences in spatial representativity between MODIS and VIIRS generated by differences in cloud screening. A possible reason for the differences in cloud contamination between VIIRS and MODIS is the use of a smoke detection test in the VIIRS product which should reduce the commission errors between smoke and cloud pixels. The consequence for the data assimilation system is that the smoke plume cannot be resolved when assimilating only MODIS data.

19. *"Line 675: again, this seems speculative and needs to be better supported with evidence or deleted. The water-leaving signal is not that large or variable at SLSTR wavelengths (green and longer), so the lack of a blue band would not be so important here. The MODIS DT ocean retrieval does not use the blue band either so it would not explain the MODIS-SLSTR differences. I don't know if the VIIRS one does, but either way, I don't think it dynamically accounts for pigment variations. So the sensors are all basically using the same spectral information, i.e. green to swIR wavelengths. I think that differences in the Southern Ocean are more likely due to different tolerances of cloud contamination and 3D effects, which may influence SLSTR differently because of the dual view and resulting parallax difference in location for clouds. See Toth (2013) for a discussion of MODIS Aqua in the Southern Ocean: https://doi.org/10.1002/jgrd.50311"*

   We agree that the NASA MODIS DT and the NOAA EPS VIIRS AOD products are not relying on the blue band to retrieve AOD over ocean. The blue band is used in the internal cloud detection schemes of both NASA MODIS and NOAA VIIRS products and for the NOAA product in the heavy aerosol identification test. We agree that the uncertainties in the retrievals and the differences between satellite products in the South ocean are likely related to cloud contamination. This is discussed in Section 5.7 where Toth et al., 2013 reference is already quoted. We thus decided to remove the statement on line 675 identified by the reviewer.

20. *"Line 689: Sirish reference should be Uprety (Sirish is the given name, Uprety the surname)."*

   We corrected the reference

21. *"Line 710: I don't think "exploits" is the right word here. Rather, the MODIS retrieval LUT contains nodes at those wind speeds."*

   We replaced "exploits" by the reviewer suggestion.

22. *"Line 748: that Sayer (2018) reference is to the NASA VIIRS data products, not the NOAA VIIRS data products used in this study, so may not be directly relevant to this point (other than to show another algorithm as a point of comparison). It was not clear to me reading whether this was intended to be an example of surface influence or an attempt to explain the results of the present study."*

   The Sayer et al., 2018 reference to the NASA VIIRS product was used as a point of comparison to illustrate the impact of uncertainties in aerosol and surface reflectance models on AOD retrieval. We agree that this can be confusing since the NASA VIIRS product is different than the NOAA VIIRS product investigated in this work. We decided to remove this statement and to keep the Tao et al., 2017 reference to support the underestimation of the MODIS DB over desert regions.

23. *"Section 6: as written, I did not find this useful as there was a lot of repetition with the immediately preceding Section 5. I suggest this is shortened and rewritten to focus more on what the abstract says the paper is about: evaluating the SLSTR and VIIRS data sets. I understand the actual assimilation will be analyzed in a follow up paper. But I think that the Conclusion here should maybe present a few brief expectations of how useful the data may/may not be, instead of repeating the previous discussion. For example, my guess is that the SLSTR product might not be useful to assimilate over ocean as it seems to be unusually low compared to the other ones. VIIRS on the other hand might be ok from NOAA20 but not SNPP, because SNPP seems to have radiometric calibration issues. So my (non modeler) takeaway from the results presented is that neither of these products are likely to help CAMS much, at least so long as the current MODIS products remain available. Is that a fair assessment, and if not, why not? This is the sort of content I think the conclusions should be giving, i.e. don't repeat the results of the analysis but more talk about what they mean. There is some of that in the current Conclusions but not much."*

We shortened the conclusion and provided recommendations for the assimilation of the investigated products based on the outcomes of this work: The assimilation of the SLSTR collection 1 product is not envisaged due to the differences in spatial representativity which are related to the stringent cloud filtering applied to the SLSTR radiances used to retrieve AOD. EUMETSAT is currently preparing a collection 3 based on a new cloud mask that will be evaluated in a future work. This paper highlights the overall good consistency between the NOAA VIIRS product and the NASA MODIS product. The consistency between the NASA MODIS and the NOAA EPS VIIRS AOD products reported in this paper shows that the assimilation of VIIRS will ensure the continuity of the CAMS data assimilation system and it will strengthen the resilience against possible future failure of MODIS. This work shows that the NOAA VIIRS product will enhance the spatial coverage of AOD observations and will provide a more accurate detection of smoke plumes. However, the conclusions reported in this paper are not sufficient to automatically include the additional AOD observations into the CAMS system and further assimilations tests are planned and will be reported on in a follow-up paper. For example, there is a need to understand how the differences between MODIS and VIIRS over ocean and land will impact the analysis. While the magnitude of the mean deviation between the products is smaller over ocean than over land, given the low AOD value of the ocean background, a slight difference in AOD between products will have a large impact on data assimilation. Since VIIRS has lower values than MODIS over ocean, its assimilation will probably decrease the analysis increment over ocean which is known to be too high due to the positive offset of TERRA/MODIS. Over land, the larger VIIRS AOD for biomass burning and dust source regions should lead to larger analysis increments that may affect AOD and surface particle matter predictions other these regions. In addition, our study reveals significant departures between products retrieved from the same instrument but from different satellite platforms and this will affect how bias correction is carried out within the system. This work shows that it would be preferable to use NOAA20/VIIRS as an anchor and apply the bias correction to S-NPP/VIIRS which was found to be positively biased over ocean. Our results also highlight the role of geometry in retrieval uncertainties that can lead to systematic differences between products. Adding the scattering angle in the current variational bias correction scheme implemented in the CAMS data assimilation system could help to represent any geometry-dependent biases in the retrieval. Finally, the observation error is an important variable to weight the relative contribution of each satellite observation to the analysis. Further work is required to evaluate the retrieval error associated with each product which could be inflated to better reflect the larger diversity between products reported in the South ocean and over bright land surfaces.

24. *"Appendix A: thank you for moving this section out of the paper into an Appendix, it makes the main paper more readable, and now if someone wants to know more details but not read the algorithm/validation papers this gives a summary."*

25. *"Appendix B: it's not obvious to me what the blue lines on the plots represent, what is it? I'm not sure they are useful and maybe they can be deleted to reduce clutter. I also think it's more useful to show the 1:1 line than what I guess is a regression line here (again, the Appendix doesn't say). That way one can more directly see whether one data set is higher/lower than another by whether they are above/below this line, without having to cross-reference the existing lines to the labels on both axes. The regression lines seem skewed by offsets at low-AOD conditions (as most of the points are there) whereas as a reader I am more interested in whether one is lower or higher than the other across the full range of AODs. That is less clear when showing the regression line than showing the 1:1 line would be."*

We modified the scatterplots and we replaced the regression line by 1:1 line.

**B.2 Response to reviewer 3**

We first re-state the objective of our work and then we address the specific points raised by the reviewer.

**1/ Clarification of the scope of the paper**

Biases in assimilated observations and departure between distinct satellite products used in the assimilation system can negatively affect data assimilation outputs (Zhang and Reid, 2005). Therefore, systematic and random differences between satellite products and departures with their model-simulated equivalents need to be properly evaluated at the model spatial resolution in order to account for them in the assimilation process (Dee et al., 2005). The objective of this work is to evaluate two new near real time (NRT) satellite AOD products to prepare their assimilation in the CAMS data assimilation system, namely the Copernicus SLSTR AOD (C1) from Sentinel-3A&B over ocean and the NOAA EPS VIIRS AOD (v2r1) from S-NPP and NOAA-20 over both land and ocean. The consistency between MODIS (C6.1), PMAp (v2.1), VIIRS (v2.r1) and SLSTR (C1) AOD products as well as their differences with the modelled AOD were monitored over a 6-month experiment, from December 2019 to May 2020. This paper aims at assessing the differences between the satellite AOD products at the model grid resolution (level-3). All analyses and conclusions reported in this paper hold for level-3 satellite AOD generated for their use in the CAMS data assimilation system and may not directly apply to level-2 retrieval at their native spatial and temporal resolution. Our work does not include 1) any intercomparison between collocated level-2 product 2) nor any evaluation against ground observations such as AERONET. Intercomparison and evaluation of level-2 satellite AOD against AERONET was achieved by data providers and other independent studies (e.g. Sogacheva, 2019; Schutgens et al., 2020). Our work complements such studies by providing unique insights on AOD product diversity at the model spatial resolution in the perspective of their use in a data assimilation system.

Our evaluation approach relies mainly on the use of the first guess departure (FGD) metric, which represents the differences between the satellite observation and its model-simulated equivalent based on the shortrange forecast. The mean and the standard deviation (SD) of FGD are meaningful estimates of the systematic and random, respectively, differences between the observation and their model-simulated equivalent which are the results of both the observation and the model errors. Any bias in the AOD retrievals can result in inconsistencies between distinct satellite observations which may fight against each other when they are assimilated, resulting in larger errors in the analysis. The use of the first guess departure is twofold: i) check that the mean departure between each satellite

observation and the model is reasonably small and not impacted by any biases in the observation and ii) evaluate the retrievals relatively to the model to identify possible spatial and temporal inconsistencies between satellite products that would impact the assimilation of multi-satellite AODs. This requires that the model is skilful to some extent and lowly biased compared to the observation. This is a reasonable assumption given that the short-term forecast used to compute FGD is simulated from an optimal estimate of the atmospheric state produced by the data assimilation system.

We clarified the objectives of the work in Introduction in the new version of the MS. We also emphasized our approach based on the evaluation of the departure with the model-simulated AOD in Section 3.3. Finally, we included a new paragraph in Conclusion to discuss recommendations for the future assimilation of the investigated products based on the outcomes of this work.

With respect to the investigated product, the VIIRS product used in this work is the EPS dataset produced by NOAA in NRT. VIIRS AOD datasets are also produced in NRT by NASA (Sayer et al., 2017; Hsu et al., 2019; Sawyer et al., 2020). However, the NASA dataset only include retrievals from S-NPP while the NOAA EPS product includes both S-NPP and NOAA20 retrievals. For this reason, it was decided to implement and test the NOAA product in CAMS. Most past intercomparison and evaluation studies concern the NASA VIIRS product and to our knowledge there has been no published intercomparison including the NOAA product which limits the use of past results for the discussion.

**2/Specific remarks**

- *"Please spell out the new terms, e.g., PMAp and SLSTR, in the Abstract, and double-check and address such issues throughout the paper."*

  We included the definition of PMAp and SLSTR acronyms in the abstract and the text.

- *"The authors are suggested to summarize the previous studies focusing on MODIS and VIIRS AOD validation and comparison in the Introduction since a large number of related studies have been carried out."*

We expanded the paragraph on past intercomparison studies (lines 66-74) by adding the following statements on past evaluation of MODIS and VIIRS AOD, both produced by NASA:

Over ocean, AOD products exhibit low correlation at low AOD which can be due to different retrieval sensitivities to errors in surface reflectance and cloud contamination (Sayer et al,., 2018; Sogacheva et al., 2020). Sayer et al., 2017 showed that intercalibrating S-NPP/VIIRS with AQUA/MODIS reduces the discrepancies in AOD between both products and improves the accuracy of VIIRS AOD over ocean. TERRA/MODIS dark target AOD, which has a positive offset over ocean related to the calibration degradation of the blue band (Levy et al., 2013), was shown to be frequently the highest of AOD products over open ocean conditions (Zhang et al., 2017, de Leeuw 2018, Sogacheva et al., 2020, Sawyer et al., 2020). Over land, product agree better over dark vegetated regions (e.g. Europe) than over bright surfaces (e.g. Africa). Larger departures for high AOD values are found for regions and periods which are dominated by anthropogenic pollution, biomass burning events or dust outbreaks (Tao et al., 2017; Sogacheva et al., 2020). Sayer et al., 2019 reported larger VIIRS Deep Blue (DB) AOD than MODIS DB over dust source regions (e.g. Sahara, Arabian Peninsula, and Taklamakan desert) that the authors relate to the improved aerosol and surface reflectance models used in the VIIRS DB algorithm.

- "Suggest adding related references for each AOD product in the Table 1."

  We added key product references in Table 1

- *"Information on AERONET measurements is missing. What version and level of AERONET data used in the current study and how to obtain the measurements at 550nm?"*

As explained in Section 1 we have not used any AERONET data in this work. The objective is to evaluate the consistency between AOD products within the IFS data assimilation system.

- *"My major concern is the study period since the authors only chose a Two 3-month period including DJF and MAM. In fact, we know that AOD shows strong seasonal cycles, which are much more obvious in regional scales, e.g., East Asia or Western U.S. What are the results in summer and autumn for the northern hemisphere? It is suggested to extend the study period (at least one year) to make the results more convincing."*

We agree that AOD has a strong seasonal and regional dependency particularly with respect to biomass burning events and dust outbreaks which influences the diversity of AOD products.

This work involves NWP model numerical simulation which requires important computing resources. The 6-month experiments used in this work represents 3 months of simulation time. Although this period misses the complete seasonal cycle, it covers winter and spring seasons which convey a significant signal in terms of differences between AOD products as evaluated from a data assimilation perspective.

Besides, ECMWF is currently moving its supercomputing resources (change of supercomputer and changes of location) which limits the use of the supercomputer in the next few months and prevents from expanding the period of study of this work.

A companion paper will present results of the assimilation of some of these products, and the study period will cover summer and autumn seasons. However, this dataset is not ready to be included in this present study.

Section 5.7 provides a discussion on the seasonal and spatial dependence of AOD retrievals and acknowledge that the study period should be extended in future work to address the seasonal and regional dependence of AOD product diversity. We emphasized this aspect in Conclusion (last paragraph) of the revised manuscript.

- *"Line 245: mean deviation (MD)? Do you mean bias?"*

  We mean differences between satellite AOD retrieval at the model spatial resolution. This is computed over the globe and the 3-month time period. There is no reference to independent ground dataset in this work to evaluate biases in satellite AOD.

- *"Tables 2-3: Suggest showing the defined ROIs in a Figure to make readers clearer."*

We added a map displaying the ROI over ocean and land (new Figure 1)

- *"Sections 4.1 and 4.2: In addition to simple descriptions on intuitive results, readers prefer to see the reasons for the differences among different aerosol products.*

We chose to have a dedicated section on the description of the results in Section 4 which helps to better understand the information conveyed by each Figure and tables. These results are used in Section 5 to discuss the reasons for the differences among satellite AOD products.

- *The authors need to focus on analyzing these regions with large differences, also, relevant literature support is needed."*

We expanded Section 5.7 to emphasize the region-dependency of the differences between satellite products. Regarding, literature support: the EPS VIIRS dataset produced NOAA, the SLSTR product and the PMAP dataset have not been included in any published intercomparison studies that limits the use of past results to discuss the results of this work.

- *"Besides mean maps, I would also like to see the differences in spatial coverage among different aerosol products, especially over land (e.g., bright surfaces)."*

Differences in instrument geometry (pixel size, swath, pixel growth along the swath) and cloud filtering can lead to differences in spatial coverage between level-2 retrievals. For example, the finer spatial resolution, at which VIIRS retrieval is performed, and its larger swath, imply more frequent retrievals compared to MODIS and explain the larger spatial coverage of VIIRS over Northern latitudes, Central Africa and South America (Sayer et al., 2019).
However, in this work, the differences in the satellite AOD products are evaluated at the model grid spatial resolution (~40 km) and not at the native resolution (level-2) of the product. At such coarse resolution, the differences in spatial coverage between retrievals are reduced (Schutgens et al., 2016,2019). But differences in cloud detection can lead to substantial differences in spatial representativity at the model grid spatial resolution between products which can generate larger differences in AOD than the differences between collocated level-2 retrievals. This is particularly illustrated in section 5.1 by the differences between SLSTR AOD values and the rest of the satellite products due to the too stringent cloud mask applied to SLSTR radiances. Differences in product spatial representativity at the model grid spatial resolution are region dependent. They will be larger over cloudy regions (e.g. mid-latitude oceanic regions), bright surfaces and high AOD regions where the commission errors with cloud detection are larger. These aspects are emphasized in the discussion in Section 5.1, 5.2 and 5.7.

- *"Also, the validation and comparison in performance over different underlying surfaces (surface brightness) are also interesting."*

As mentioned above, this study does not include any validation results against AERONET. Section 5.7 already contains a discussion on the larger uncertainties in AOD retrieval over bright surfaces.

Sincerely,

Sebastien Garrigues on behalf of co-authors

---

## Author Response (AR2)

Dr Sebastien Garrigues
Copernicus Atmosphere Monitoring Service (CAMS)
European Centre for Medium-Range Weather Forecasts (ECMWF)
Shinfield Park, Reading, RG2 9AX, United Kingdom
E-mail: sebastien.garrigues@ecmwf.int
Tel: +44 (0) 118 949 9264

Reading, 22/08/2022,

Answer to final review (technical corrections) of "Monitoring multiple satellite Aerosol Optical Depth (AOD) products within the Copernicus Atmosphere Monitoring Service (CAMS) data assimilation system" as a research paper for ACP.

Dear Editor,

Please find the revised version of the manuscript entitled "Monitoring multiple satellite Aerosol Optical Depth (AOD) products within the Copernicus Atmosphere Monitoring Service (CAMS) data assimilation system" along with the response to reviewer 3 technical corrections.

We would like to thank reviewer 3 for the suggested modifications on the paper that helped to improve the quality of the paper.

1) *Some related work focusing on evaluation of various global aerosol products should be mentioned, e.g.,*
*https://doi.org/10.5194/acp-19-7183-2019*
*https://doi.org/10.1016/j.atmosenv.2018.12.004*

⇨ We added both references. We used them in Introduction (line 72, line 80-82) and in the discussion (line 784)

2) *Line 145: 201?*

⇨ We corrected the typo, replaced by Sayer et al., 2018a

3) *Figures 2, 3, 6, 7: The interval between the upper and lower figures is too large. Better add (a), …, (d) for each subplot (similar issue for other figures).*

=>The interval between the upper and lower panels has been removed. We chose to not add a), b) … to identify the panels because each panel has an explicit title that clearly indicates the AOD product.

4) *Too many figures for a research paper, any possibility for merging some graphs (as subplots) of the same type, e.g., Figures 4 and 5; 8 and 9; 15 and 16; and 17 and 18?*

⇨ We reduced the number of figures by grouping 6 and 7; 8 and 9; 15 and 16; 17 and 18 as suggested.

Sincerely,

Sebastien Garrigues on behalf of co-authors